# Deep Repulsive Clustering of Ordered Data Based on Order-Identity Decomposition

**Seon-Ho Lee and Chang-Su Kim**
School of Electrical Engineering, Korea University
`seonholee@mcl.korea.ac.kr, changsukim@korea.ac.kr`

## Abstract

We propose the deep repulsive clustering (DRC) algorithm of ordered data for effective order learning. First, we develop the order-identity decomposition (ORID) network to divide the information of an object instance into an order-related feature and an identity feature. Then, we group object instances into clusters according to their identity features using a repulsive term. Moreover, we estimate the rank of a test instance, by comparing it with references within the same cluster. Experimental results on facial age estimation, aesthetic score regression, and historical color image classification show that the proposed algorithm can cluster ordered data effectively and also yield excellent rank estimation performance.

## 1 Introduction

There are various types of 'ordered' data. For instance, in facial age estimation (Ricanek & Tesafaye, 2006), face photos are ranked according to the ages. Also, in a video-sharing platform, videos can be sorted according to the numbers of views or likes. In these ordered data, classes, representing ranks or preferences, form an ordered set (Schröder, 2003). Attempts have been made to estimate the classes of objects, including multi-class classification (Pan et al., 2018), ordinal regression (Frank & Hall, 2001), metric regression (Fu & Huang, 2008). Recently, a new approach, called order learning (Lim et al., 2020), was proposed to solve this problem.

Order learning is based on the idea that it is easier to predict ordering relationship between objects than to estimate the absolute classes (or ranks); telling the older one between two people is easier than estimating their exact ages. Hence, in order learning, the pairwise ordering relationship is learned from training data. Then, the rank of a test object is estimated by comparing it with reference objects with known ranks. However, some objects cannot be easily compared. It is less easy to tell the older one between people of different genders than between those of the same gender. Lim et al. (2020) tried to deal with this issue, by dividing an ordered dataset into disjoint chains. But, the chains were not clearly separated, and no meaningful properties were discovered from the chains.

In this paper, we propose a reliable clustering algorithm, called deep repulsive clustering (DRC), of ordered data based on order-identity decomposition (ORID). Figure 1 shows a clustering example of ordered data. Note that some characteristics of objects, such as genders or races in age estimation, are not related to their ranks, and the ranks of objects sharing such characteristics can be compared more reliably. To discover such characteristics without any supervision, the proposed ORID network decomposes the information of an object instance into an order-related feature and an identity feature unrelated to the rank. Then, the proposed DRC clusters object instances using their identity features; in each cluster, the instances share similar identity features. Furthermore, given a test instance, we decide its cluster based on the nearest neighbor (NN) rule, and compare it with reference instances within the cluster to estimate its rank. To this end, we develop a maximum *a posteriori* (MAP) estimation rule. Experimental results on ordered data for facial age estimation, aesthetic score regression (Kong et al., 2016), and historical color image classification (Palermo et al., 2012) demonstrate that the proposed algorithm separates ordered data clearly into meaningful clusters and provides excellent rank estimation performances for unseen test instances.

The contributions of this paper can be summarized as follows.

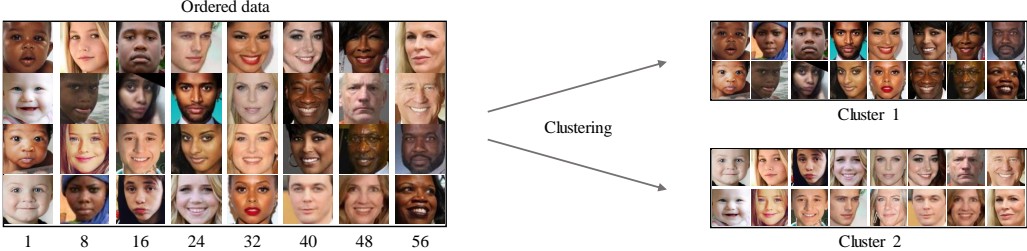

Figure 1: A clustering example of facial photos, which are ordered according to ages. Without any supervision, the proposed algorithm can obtain meaningful clusters using identity features.

- We first propose the notion of identity features of ordered data and develop the ORID network for the order-identity decomposition.
- We develop the DRC algorithm to cluster data on a unit sphere effectively using a repulsive term. We also prove the local optimality of the solution.
- We propose the MAP decision rule for rank estimation. The proposed algorithm provides the state-of-the-art performances for facial age estimation and aesthetic score regression.

## 2 RELATED WORK

### 2.1 ORDER LEARNING

The notion of order learning was first proposed by Lim et al. (2020). It aims to determine the order graph of classes and classify an object into one of the classes. In practice, it trains a pairwise comparator, which is a ternary classifier, to categorize the relationship between two objects into one of three cases: one object is bigger than, similar to, or smaller than the other. Then, it estimates the rank of a test object, by comparing it with reference objects with known ranks. However, not every pair of objects are easily comparable. Although Lim et al. (2020) attempted to group objects into clusters, in which objects could be more accurately compared, their clustering results were unreliable.

Pairwise comparison has been used to estimate object ranks, because relative evaluation is easier than absolute evaluation in general. Saaty (1977) proposed the scaling method to estimate absolute priorities from relative priorities, which has been applied to various decision processes, including aesthetic score regression (Lee & Kim, 2019). Also, some learning to rank (LTR) algorithms are based on pairwise comparison (Liu, 2009; Cohen et al., 1998; Burges et al., 2005; Tsai et al., 2007).

Order learning attempts to combine (possibly inconsistent) pairwise ordering results to determine the rank of each object. Thus, it is closely related to the Cohen et al.'s LTR algorithm (1998), which learns a pairwise preference function and obtains a total order of a set to maximize agreements among preference judgments of pairs of elements. Also, order learning is related to rank aggregation (Dwork et al., 2001), in which partially ordered sets are combined into a linearly ordered set to achieve the maximum consensus among those partial sets. Rank aggregation has been studied in various fields (Brüggemann et al., 2004). Since optimal aggregation is NP-hard, Dwork et al. (2001) proposed an approximate algorithm, called Markov chain ordering. There are many other approximate schemes, such as the local Kemenization, Borda count, and scaled footrule aggregation.

### 2.2 CLUSTERING

Data clustering is a fundamental problem to partition data into disjoint groups, such that elements in the same group are similar to one another but elements from different groups are dissimilar. Although various clustering algorithms have been proposed (Hartigan & Wong, 1979; Ester et al., 1996; Kohonen, 1990; Dhillon & Modha, 2001; Reynolds, 2009), conventional algorithms often yield poor performance on high-dimensional data due to the curse of dimensionality and ineffectiveness of similarity metrics. Dimensionality reduction and feature transform methods have been studied to map raw data into a new feature space, in which they are more easily separated. Linear transforms, such as PCA (Wold et al., 1987), and non-linear transformations, including kernel methods (Hofmann et al., 2008) and spectral clustering (Ng et al., 2002), have been proposed.

Recently, deep neural networks have been adopted effectively as feature embedding functions (LeCun et al., 2015), and these deep-learning-based feature embedding functions have been combined with classical clustering algorithms. For instance, Caron et al. (2018) proposed a deep clustering algorithm based on $k$-means. It clusters features from a neural network and then trains the network using the cluster assignments as pseudo-labels. This is done iteratively. Also, Yang et al. (2016) jointly learned feature representations and clustered images, based on agglomerative clustering. Chang et al. (2017) recast the image clustering task into a binary classification problem to predict whether a pair of images belong to the same cluster or different clusters. Similarly to these algorithms, we use a neural network to determine a feature space in which clustering is done more effectively. However, we consider the clustering of ordered data, and each cluster should consists of elements, whose ranks can be compared more accurately.

There are conventional approaches to use clustering ideas to aid in classification or rank estimation. For example, Yan et al. (2015) developed a hierarchical classifier, which clusters fine categories into coarse category groups and classifies an object into a fine category within its coarse category group. For extreme multiclass classification, Daumé III et al. (2017) proposed to predict a class label among candidate classes only, which are dynamically selected by the recall tree. It is however noted that the leaves of the recall tree do not partition the set of classes. Also, for age estimation, Li et al. (2019) proposed a tree-like structure, called bridge-tree, to divide data into overlapping age groups and train a local regressor for each group. The set of local regressors can be more accurate than a global regressor to deal with the entire age range. Whereas these conventional approaches group data in the label dimension to perform their tasks more effectively, the proposed algorithm cluster data in the dimension orthogonal to the label dimension. In other words, we cluster data using identity features, instead of using order features.

## 3 PROPOSED ALGORITHM

### 3.1 PROBLEM DEFINITION

An *order* is a binary relation, often denoted by $\leq$, on a set $\Theta = \{\theta_1, \theta_2, \ldots, \theta_m\}$ (Schröder, 2003). It should satisfy three properties of reflexivity ($\theta_i \leq \theta_i$ for all $i$), antisymmetry ($\theta_i \leq \theta_j$ and $\theta_j \leq \theta_i$ imply $\theta_i = \theta_j$), and transitivity ($\theta_i \leq \theta_j$ and $\theta_j \leq \theta_k$ imply $\theta_i \leq \theta_k$). Then, $\Theta$ is called a *partially ordered set*. Furthermore, if every pair of elements are comparable ($\theta_i \leq \theta_j$ or $\theta_j \leq \theta_i$ for all $i, j$), $\Theta$ is called a *chain* or *linearly ordered set*.

An order describes ranks or priorities of classes. For example, in age estimation, $\theta_i$ may represent the age class of $i$-year-olds. Then, $\theta_{14} \leq \theta_{49}$ represents that 14-year-olds are younger than 49-year-olds. As mentioned previously, it is less easy to tell the older one between people of different genders. An algorithm, hence, may compare a subject with reference subjects of the same gender only. In such a case, each age class $\theta_i$ represents two subclasses $\theta_i^{\text{female}}$ and $\theta_i^{\text{male}}$ of different types, and the algorithm compares only subjects of the same type. Lim et al. (2020) assumed that subclasses of different types are incomparable and thus the set of subclasses is the union of $k$ disjoint chains, where $k$ is the number of types. However, in many ranking applications, objects of different types can be compared (although less easily than those of the same type are). Thus, instead of assuming incomparability across chains, we assume that there is a total order on $\Theta = \{\theta_1, \theta_2, \ldots, \theta_m\}$, in which each class $\theta_i$ consists of $k$ types of subclasses, and that object instances of the same type are more easily compared than those of different types.

Suppose that $n$ training instances in $\mathcal{X} = \{x_1, x_2, \ldots, x_n\}$ are given. Also, suppose that there are $m$ ranks and the ground-truth rank of each instance is known. In this sense, $\mathcal{X}$ contains ordered data. The problem is twofold. The first goal is to decompose the whole instances $\mathcal{X}$ into $k$ disjoint clusters $\{\mathcal{C}_j\}_{j=1}^k$ in which instances are more easily compared;

$$\mathcal{X} = \bigcup_{j=1}^k \mathcal{C}_j \tag{1}$$

where $\mathcal{C}_i \cap \mathcal{C}_j = \emptyset$ for $i \neq j$. In other words, we aim to partition the ordered data in $\mathcal{X}$ into $k$ clusters, by grouping them according to their characteristics unrelated to their ranks. These characteristics, which tend to remain the same even when an object experiences rank changes, are referred to as 'identity' features in this work. For example, in age estimation, genders or races can be identity features. However, we perform the clustering without any supervision for identity features. Notice

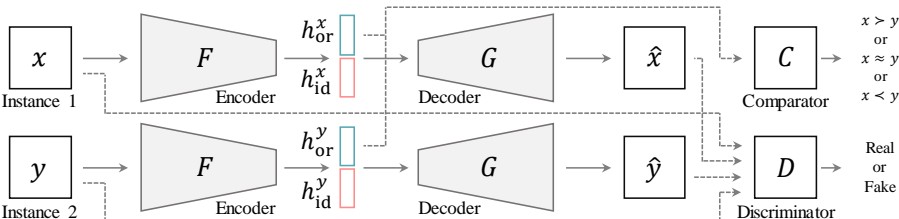

Figure 2: An overview of the ORID network.

that instances within a cluster would be compared more easily than those across clusters, since they have similar identity features. The number $k$ of clusters is assumed to be known *a priori*. Impacts of $k$ on the clustering performance are discussed in Appendix B.7. The second goal is to assign an unseen test instance into one of the clusters and determine its rank by comparing it with reference instances within the cluster. To achieve these goals, we propose the ORID network and the DRC algorithm.

### 3.2   ORDER-IDENTITY DECOMPOSITION

In general, object instances can be compared more easily, as they have more similar identity features irrelevant to order. Therefore, we decompose the information of each object instance into an order feature and an identity feature. To this end, we propose the ORID network in Figure 2, composed of three parts: autoencoder, discriminator, and comparator.

1) Autoencoder: Similarly to deep clustering algorithms in (Yang et al., 2017; Dizaji et al., 2017; Chen et al., 2017; Ji et al., 2017), we use the autoencoder $G \circ F(\cdot)$, based on a neural network, to extract feature vectors. The encoder $h^x = F(x)$ maps an input vector $x$ to a feature vector $h^x$, while the decoder $\hat{x} = G(h^x)$ reconstructs $\hat{x}$ from $h^x$. By minimizing the reconstruction loss $\|x - \hat{x}\|_1$, $F$ is trained to represent $x$ compactly with as little loss of information as possible. We decompose the overall feature $h^x \in \mathbb{R}^{d_{or}+d_{id}}$ into the order feature $h^x_{or}$ and the identity feature $h^x_{id}$, given by

$$h^x_{or} = [h^x_1, h^x_2, \ldots, h^x_{d_{or}}]^t \tag{2}$$

$$h^x_{id} = [h^x_{d_{or}+1}, h^x_{d_{or}+2}, \ldots, h^x_{d_{or}+d_{id}}]^t / \|[h^x_{d_{or}+1}, h^x_{d_{or}+2}, \ldots, h^x_{d_{or}+d_{id}}]\| \tag{3}$$

where $d_{or}$ and $d_{id}$ are the dimensions of $h^x_{or}$ and $h^x_{id}$. However, without additional control, the output $h^x$ of the neural network $F$ would be highly entangled (Higgins et al., 2018). To put together order-related information into $h^x_{or}$, we employ the comparator.

2) Comparator: Using the order features $h^x_{or}$ and $h^y_{or}$ of a pair of instances $x$ and $y$, we train the comparator, which classifies their ordering relationship into one of three categories 'bigger,' 'similar,' and 'smaller':

$$x \succ y \text{ if } \theta(x) - \theta(y) > \tau, \quad x \approx y \text{ if } |\theta(x) - \theta(y)| \leq \tau, \quad x \prec y \text{ if } \theta(x) - \theta(y) < -\tau, \tag{4}$$

where $\theta(\cdot)$ denotes the class of an instance. As in (Lim et al., 2020), '$\succ, \approx, \prec$' represent the ordering relationship between instances, while '$>, =, <$' do the mathematical order between classes. The comparator outputs the softmax probability $p^{xy} = (p^{xy}_\succ, p^{xy}_\approx, p^{xy}_\prec)$. It is trained to minimize the cross-entropy between $p^{xy}$ and the ground-truth one-hot vector $q^{xy} = (q^{xy}_\succ, q^{xy}_\approx, q^{xy}_\prec)$. Because it is trained jointly with the autoencoder, the information deciding the ordering relationship tends to be encoded into the order features $h^x_{or}$ and $h^y_{or}$. On the other hand, the remaining information necessary for the reconstruction of $\hat{x}$ and $\hat{y}$ are encoded into the identity features $h^x_{id}$ and $h^y_{id}$.

3) Discriminator: We adopt the discriminator $D$ that tells real images from synthesized images, generated by the decoder $G$. Using the GAN loss (Goodfellow et al., 2014), the discriminator helps the decoder to reconstruct more realistic output $\hat{x}$ and $\hat{y}$.

Appendix A provides detailed network structures of these components in ORID.

### 3.3   DEEP REPULSIVE CLUSTERING

After obtaining the identity features $h^{x_1}_{id}, h^{x_2}_{id}, \ldots, h^{x_n}_{id}$ of all instances $x_i \in \mathcal{X}$, we partition them into $k$ clusters. Each cluster contains instances that are more easily comparable to one another. The

identity features are normalized in Eq. (3) and lie on the unit sphere in $\mathbb{R}^{d_{\mathrm{id}}}$. In other words, we cluster data points on the unit sphere. Thus, the cosine similarity is a natural affinity metric. Let $\mathcal{C}_j$, $1 \leq j \leq k$, denote the $k$ clusters. Also, let $c_j$, constrained to be on the unit sphere, denote the 'centroid' or the representative vector for the instances in cluster $\mathcal{C}_j$. We define the quality of cluster $\mathcal{C}_j$ as

$$\sum_{x \in \mathcal{C}_j} \left( (h_{\mathrm{id}}^x)^t c_j - \alpha \frac{1}{k-1} \sum_{l \neq j} (h_{\mathrm{id}}^x)^t c_l \right) \tag{5}$$

where the first term is the similarity of an instance in $\mathcal{C}_j$ to the centroid $c_j$, the second term with the negative sign quantifies the average dissimilarity of the instance from the other centroids, and $\alpha$ is a nonnegative weight. For a high quality cluster, instances should be concentrated around the centroid and be far from the other clusters. The second term is referred to as the repulsive term, as its objective is similar to the repulsive rule in (Lee et al., 2015). Although conventional methods also try to increase inter-cluster dissimilarity (Ward Jr, 1963; Lee et al., 2015), to the best of our knowledge, DRC is the first attempt to use an explicit repulsive term in deep clustering, which jointly optimizes clustering and feature embedding. Next, we measure the overall quality of the clustering by

$$J(\{\mathcal{C}_j\}_{j=1}^k, \{c_j\}_{j=1}^k) = \sum_{j=1}^k \sum_{x \in \mathcal{C}_j} \left( (h_{\mathrm{id}}^x)^t c_j - \alpha \frac{1}{k-1} \sum_{l \neq j} (h_{\mathrm{id}}^x)^t c_l \right). \tag{6}$$

We aim to find the optimum clusters to maximize this objective function $J$, yet finding the global optimum is NP-complete (Kleinberg et al., 1998; Garey et al., 1982). Hence, we propose an iterative algorithm, called DRC, to find a local optimum, as in the $k$-means algorithm (Gersho & Gray, 1991).

1. Centroid rule: After fixing the clusters $\{\mathcal{C}_j\}_{j=1}^k$, we update the centroids $\{c_j\}_{j=1}^k$ to maximize $J$ in Eq. (6). Because the centroids should lie on the unit sphere, we solve the constrained optimization problem:

$$\text{maximize } J(\{c_j\}_{j=1}^k) \text{ subject to } c_j^t c_j = 1 \text{ for all } j = 1, \ldots, k. \tag{7}$$

Using Lagrangian multipliers (Bertsekas, 1996), the optimal centroids are obtained as

$$c_j = \left( \sum_{x \in \mathcal{C}_j} h_{\mathrm{id}}^x - \alpha \frac{1}{k-1} \sum_{x \in \mathcal{X} \setminus \mathcal{C}_j} h_{\mathrm{id}}^x \right) \Big/ \left\| \sum_{x \in \mathcal{C}_j} h_{\mathrm{id}}^x - \alpha \frac{1}{k-1} \sum_{x \in \mathcal{X} \setminus \mathcal{C}_j} h_{\mathrm{id}}^x \right\|. \tag{8}$$

2. NN rule: On the other hand, after fixing the centroids, we update the membership of each instance to maximize $J$ in Eq. (6). The optimal cluster $\mathcal{C}_j$ is given by

$$\mathcal{C}_j = \left\{ x \,|\, (h_{\mathrm{id}}^x)^t c_j \geq (h_{\mathrm{id}}^x)^t c_l \text{ for all } 1 \leq l \leq k \right\}. \tag{9}$$

In other words, an instance should be assigned to $\mathcal{C}_j$ if its nearest centroid is $c_j$.

We apply the centroid rule and the NN rule iteratively until convergence. Because both rules monotonically increase the same objective function $J$ and the inequality $J \leq n + \frac{\alpha}{k-1} n$ always holds, $J$ is guaranteed to converge to a local maximum. Readers interested in the convergence are referred to (Sabin & Gray, 1986; Pollard, 1982).

Without the repulsive term in Eq. (6) (*i.e.* at $\alpha = 0$), centroid $c_j$ in Eq. (8) is updated by

$$c_j = \sum_{x \in \mathcal{C}_j} h_{\mathrm{id}}^x \Big/ \left\| \sum_{x \in \mathcal{C}_j} h_{\mathrm{id}}^x \right\|, \tag{10}$$

as done in the spherical $k$-means (Dhillon & Modha, 2001). In contrast, with a positive $\alpha$, the objective function $J$ is reduced when the centroids are far from one another. Ideally, in equilibrium, the centroid of a cluster should be the opposite of the centroid of all the other clusters;

$$\left( \frac{\sum_{x \in \mathcal{C}_j} h_{\mathrm{id}}^x}{\| \sum_{x \in \mathcal{C}_j} h_{\mathrm{id}}^x \|} \right)^t \left( \frac{\sum_{x \in \mathcal{X} \setminus \mathcal{C}_j} h_{\mathrm{id}}^x}{\| \sum_{x \in \mathcal{X} \setminus \mathcal{C}_j} h_{\mathrm{id}}^x \|} \right) = -1 \quad \text{for all } j = 1, 2, \ldots, k. \tag{11}$$

Note that the ORID network and thus the encoded feature space are trained jointly with the repulsive clustering. As the training goes on, the centroids repel one another, and the clusters are separated more clearly due to the repulsive term.

We jointly optimize the clusters and the ORID network parameters, as described in **Algorithm 1**. First, we train the ORID network for warm-up epochs, by employing every pair of instances $x$ and $y$ as input. Then, using the identity features, we partition the input data into $k$ clusters using $k$-means. Second, we repeat the fine-tuning of the ORID network and the repulsive clustering alternately. In the fine-tuning, a pair of $x$ and $y$ are constrained to be from the same cluster, and the following loss function is employed.

$$\ell = \lambda_{\mathrm{rec}} \ell_{\mathrm{rec}} + \lambda_{\mathrm{clu}} \ell_{\mathrm{clu}} + \lambda_{\mathrm{com}} \ell_{\mathrm{com}} + \lambda_{\mathrm{gan}} \ell_{\mathrm{gan}}. \tag{12}$$

Appendix B describes this loss function in detail, proves the optimality of the centroid and NN rules in Eq. (8) and (9), and analyzes the impacts of the repulsive term in Eq. (6).

---

**Algorithm 1** DRC-ORID

---

**Input:** Ordered data $\mathcal{X} = \{x_1, x_2, \ldots, x_n\}$, $k$ = the number of clusters

  1: Train ORID network for warm-up epochs to minimize loss $\lambda_{\text{rec}}\ell_{\text{rec}} + \lambda_{\text{com}}\ell_{\text{com}} + \lambda_{\text{gan}}\ell_{\text{gan}}$
  2: Partition $\mathcal{X}$ into $\mathcal{C}_1, \mathcal{C}_2, \ldots, \mathcal{C}_k$ using $k$-means
  3: **repeat**
  4:      Fine-tune ORID network to minimize loss $\lambda_{\text{rec}}\ell_{\text{rec}} + \lambda_{\text{clu}}\ell_{\text{clu}} + \lambda_{\text{com}}\ell_{\text{com}} + \lambda_{\text{gan}}\ell_{\text{gan}}$
  5:      **repeat**
  6:          **for all** $j = 1, 2, \ldots, k$ **do**
  7:              Update centroid $c_j$ via Eq. (8)                          ▷ Centroid rule
  8:          **end for**
  9:          **for all** $j = 1, 2, \ldots, k$ **do**
10:              Update cluster $\mathcal{C}_j$ via Eq. (9)                         ▷ NN rule
11:          **end for**
12:      **until** convergence or predefined number of iterations
13: **until** predefined number of epochs

**Output:** Clusters $\{\mathcal{C}_j\}_{j=1}^k$, centroids $\{c_j\}_{j=1}^k$, ORID network

---

## 3.4 RANK ESTIMATION

Using the output of the DRC-ORID algorithm, we can estimate the rank of an unseen test instance $x$. First, we extract its identity feature $h_{\text{id}}^x$ using the ORID encoder. By comparing $h_{\text{id}}^x$ with the centroids $\{c_j\}_{j=1}^k$ based on the NN rule, we find the most similar centroid $c_l$. Then, $x$ is declared to belong to cluster $\mathcal{C}_l$. Without loss of generality, let us assume that the classes (or ranks) are the first $m$ natural numbers, $\Theta = \{1, 2, \ldots m\}$. Then, for each $i \in \Theta$, we select a reference instance $y_i$ with rank $i$ from cluster $\mathcal{C}_l$, so that it is the most similar to $x$. Specifically,

$$y_i = \arg\max_{y \in \mathcal{C}_l \, : \, \theta(y)=i} (h_{\text{id}}^x)^t h_{\text{id}}^y. \tag{13}$$

We estimate the rank $\theta(x)$ of the test instance $x$, by comparing it with the chosen references $y_i$, $1 \leq i \leq m$. For the rank estimation, Lim et al. (2020) developed the maximum consistency rule, which however does not exploit the probability information, generated by the comparator. In this paper, we use the maximum *a posteriori* (MAP) estimation rule, which is described in detail in Appendix B.10.

## 4 EXPERIMENTAL RESULTS

This section provides various experimental results. Due to space limitation, implementation details and more results are available in Appendices C, D, and E.

### 4.1 FACIAL AGE ESTIMATION

**Datasets:** We use two datasets. First, MORPH II (Ricanek & Tesafaye, 2006) is a collection of about 55,000 facial images in the age range $[16, 77]$. It provides gender (female, male) and race (African American, Asian, Caucasian, Hispanic) labels as well. We employ the four evaluation settings A, B, C, and D in Appendix C.2. Second, the balanced dataset (Lim et al., 2020) is sampled from the three datasets of MORPH II, AFAD (Niu et al., 2016), and UTK (Zhang et al., 2017) to overcome bias to specific ethnic groups or genders. It contains about 6,000 images for each combination of gender in {female, male} and ethnic group in {African, Asian, European}.

**Clustering:** Figure 3 shows clustering results on MORPH II (setting A), when the number of clusters is $k = 2$. Setting A contains faces of Caucasian descent only. Thus, the proposed DRC-ORID divides those faces into two clusters according to genders in general, although the annotated gender information is not used. Most males are assigned to cluster 1, while a majority of females to cluster 2. On the other hand, setting B consists of Africans and Caucasians. Thus, those images are clustered according to the races, as shown in Appendix C.3. Figure 4 is the results on the balanced dataset at $k = 3$, which is composed of MORPH II, AFAD, and UTK images. Due to different characteristics of these sources, images are clearly divided according to their sources. At $k = 2$, MORPH II images are separated from the others. This is because, unlike the MORPH II images, the boundaries of most AFAD and UTK images are zeroed for alignment using SeetaFaceEngine (Zhang et al., 2014).

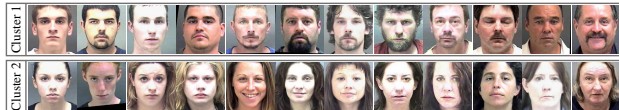 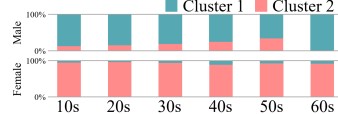

Figure 3: MORPH II images in setting A are divided into two clusters.

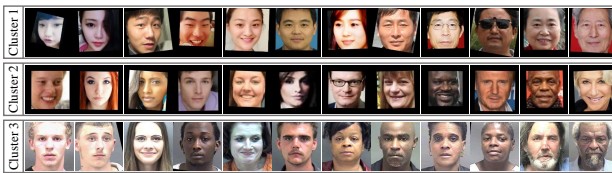 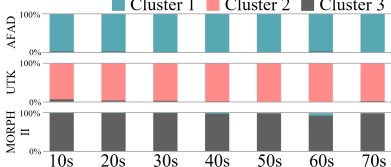

Figure 4: The balanced dataset is divided into three clusters, which are mostly composed of AFAD, UTK, and MORPH II images, respectively.

Table 1: Comparison of age estimation results on MORPH II. Here, $*$ means that the algorithm is pre-trained on the IMDB-WIKI dataset (Rothe et al., 2018).

| Algorithm | Setting A | | Setting B | | Setting C | | Setting D | |
|---|---|---|---|---|---|---|---|---|
| | MAE | CS (%) | MAE | CS (%) | MAE | CS (%) | MAE | CS (%) |
| DRFs (Shen et al., 2018) | 2.91 | 82.9 | 2.98 | - | - | - | 2.17 | 91.3 |
| MO-CNN$^*$ (Tan et al., 2017) | 2.52 | 85.0 | 2.70 | - | - | - | - | - |
| MV (Pan et al., 2018) | - | - | - | - | 2.80 | 87.0 | 2.41 | 90.0 |
| MV$^*$ (Pan et al., 2018) | - | - | - | - | 2.79 | - | 2.16 | - |
| BridgeNet$^*$ (Li et al., 2019) | 2.38 | 91.0 | 2.63 | 86.0 | - | - | - | - |
| AVDL$^*$ (Wen et al., 2020) | 2.37 | - | 2.53 | - | - | - | **1.94** | - |
| OL$^*$ (Lim et al., 2020) | 2.41 | 91.7 | 2.75 | 88.2 | 2.68 | 88.8 | 2.22 | 93.3 |
| Proposed-Vanilla ($k = 2$) | 3.36 | 80.1 | 3.32 | 80.1 | 3.40 | 79.7 | 2.99 | 84.9 |
| Proposed-VGG ($k = 1$)$^*$ | 2.35 | 92.4 | 2.64 | 88.9 | 2.64 | 89.0 | 2.22 | 93.3 |
| Proposed-VGG ($k = 2$)$^*$ | **2.26** | **93.8** | **2.51** | **89.7** | **2.58** | **89.5** | 2.16 | **93.5** |

Lim et al. (2020) also tried the clustering of the balanced dataset. Figure 5 visualizes the feature space using t-SNE (Maaten & Hinton, 2008). Although their method aligns the features according to ages, their clusters are not separated, overlapping one another. In contrast, the proposed DRC-ORID separates the three clusters clearly, as well as sorts features according to the ages within each cluster. More t-SNE plots for analyzing the impacts of the repulsive term are available in Appendix B.5.

**Age transformation:** We assess the decomposition performance of ORID. Although ORID is not designed for age transformation (Or-El et al., 2020), it decomposes an image $x$ into the order and identity features, $h_{or}^x$ and $h_{id}^x$. Thus, the age can be transformed in two steps. First, we replace $h_{or}^x$ of $x$ with $h_{or}^y$ of a reference image $y$ at a target age. Second, we decode the resultant feature (concatenation of $h_{or}^y$ and $h_{id}^x$) to obtain the transformed image. Figure 6 shows some results on MORPH II images. Order-related properties, such as skin textures and hair colors, are modified plausibly, but identity information is preserved. This indicates the reliability of ORID.

**Age estimation:** Table 1 compares the proposed algorithm with conventional age estimators on the four evaluation settings of MORPH II. These conventional algorithms take $224 \times 224$ or bigger images as input, while ORID takes $64 \times 64$ images. Moreover, most of them adopt VGG16 (Simonyan & Zisserman, 2015) as their backbones, which is more complicated than the ORID encoder. Thus, for comparison, after fixing clusters using DRC-ORID, we train another pairwise comparator based on VGG16, whose architecture is the same as Lim et al. (2020). We measure the age estimation performance by the mean absolute error (MAE) and the cumulative score (CS). MAE is the average absolute error between estimated and ground-truth ages, and CS computes the percentage of test samples whose absolute errors are less than or equal to a tolerance level of 5.

Mainly due to the smaller input size of $64 \times 64$, the vanilla version yields poorer performances than the conventional algorithms. The VGG version, however, outperforms them significantly. First, in the proposed-VGG ($k = 1$), all instances can be compared, as in the OL algorithm. In other words, the clustering is not performed. Thus, the pairwise comparators of OL and the proposed-VGG ($k = 1$) are trained in the same way, but their rank estimation rules are different. Whereas

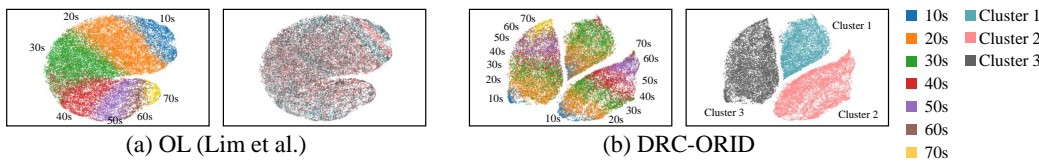

Figure 5: t-SNE visualization of the feature spaces of the balanced dataset at $k = 3$.

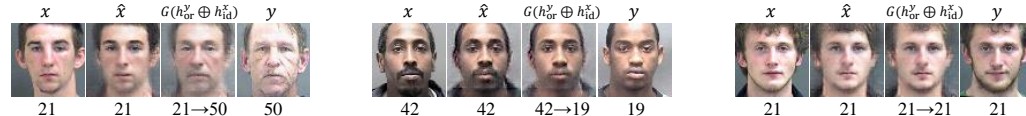

Figure 6: Age transformation results. For each test, the input $x$, reconstruction $\hat{x} = G \circ F(x)$, transformed result $G(h_{or}^y \oplus h_{id}^x)$, and reference $y$ are shown, where $\oplus$ denotes concatenation.

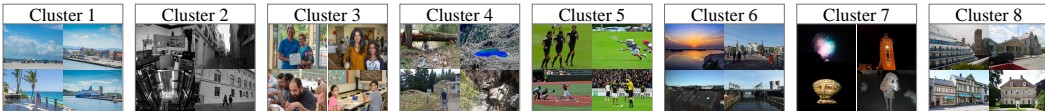

Figure 8: Example AADB images grouped into eight clusters ($k = 8$).

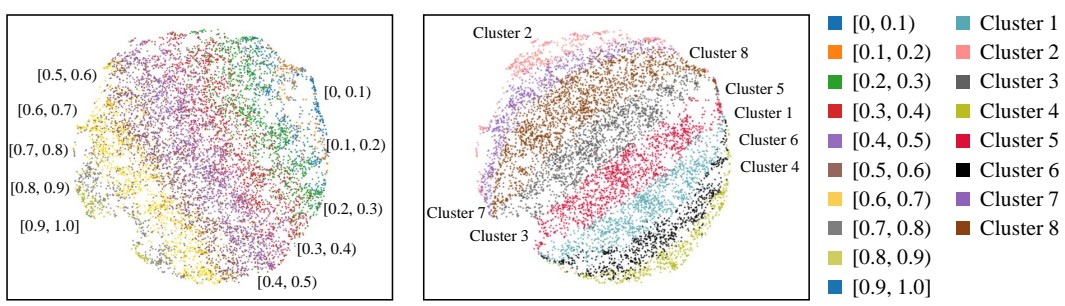

Figure 9: t-SNE visualization of feature space of AADB at $k = 8$.

OL uses the maximum consistency rule, the proposed algorithm performs the MAP estimation. The score gaps between them confirm that the MAP estimation is more accurate. Moreover, by clustering facial images into two groups, the proposed-VGG ($k = 2$) improves the results meaningfully. The proposed-VGG ($k = 2$) provides the state-of-the-art results, except for the MAE test in setting D.

## 4.2 AESTHETIC SCORE REGRESSION

The aesthetics and attribute database (AADB) is composed of 10,000 photographs of various themes such as scenery and close-up (Kong et al., 2016). Each image is annotated with an aesthetic score in $[0, 1]$. We quantize the continuous score with a step size of 0.01 to make 101 score classes. Compared to facial images, AADB contains more diverse data. It is hence more challenging to cluster AADB images. Figure 8 shows example images in each cluster at $k = 8$. Images in the same cluster have similar colors, similar contents, or similar composition. This means that ORID extracts identity features effectively, corresponding to contents or styles that are not directly related to aesthetic scores. Using those identity features, DRC discovers meaningful clusters. Figure 9 visualizes the feature space of AADB. Aesthetic scores are sorted along one direction, while clusters are separated in the other orthogonal direction. In other words, the scores look like latitudes, while the clusters appear to be separated by meridians (or lines of longitude). As a point on the earth surface can be located by its latitude and longitude, an image is represented by its aesthetic score (order feature) and cluster (identity feature).

Table 2 compares regression results. Even without clustering process, the proposed algorithm outperforms the Reg-Net and ASM algorithms. Moreover, by using the eight unsupervised clusters in Figure 8, the proposed algorithm further reduces the MAE to yield the state-of-the-art result.

Table 2: Aesthetic score regression performances of the proposed algorithm and the conventional Reg-Net (Kong et al., 2016) and ASM (Lee & Kim, 2019) on the AADB dataset.

| Algorithm | Reg-Net | ASM | Proposed ($k = 1$) | Proposed ($k = 8$) |
|---|---|---|---|---|
| MAE | 0.1268 | 0.1141 | 0.1109 | **0.1056** |

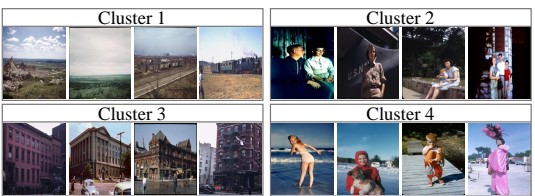

Figure 7: Example HCI images grouped into four clusters ($k = 4$).

Table 3: Comparison of classification performances on the HCI dataset.

| Model | Accuracy (%) | MAE (decade) |
|---|---|---|
| Frank & Hall (2001) | 41.36 | 0.99 |
| Cardoso & da Costa (2007) | 41.31 | 0.95 |
| Palermo et al. (2012) | 44.92 | 0.93 |
| RED-SVM (Lin & Li, 2012) | 35.92 | 0.96 |
| ORCNN (Niu et al., 2016) | 44.67 | 0.81 |
| CNNPOR (Liu et al., 2018) | **50.12** | 0.82 |
| GP-DNNOR (Liu et al., 2019) | 46.60 | **0.76** |
| Proposed ($k = 4$) | 44.72 | 0.80 |

## 4.3 HISTORICAL COLOR IMAGE CLASSIFICATION

HCI (Palermo et al., 2012) is a dataset for determining the decade when a photograph was taken. It contains images from five decades from 1930s to 1970s. Each decade category has 265 images: 210, 5, and 50 are used for training, validation and testing. Figure 7 shows the clustering results at $k = 4$. We observe similarity of contents in each cluster. Table 3 compares the quinary classification results. Frank & Hall (2001), Cardoso & da Costa (2007), Palermo et al. (2012), and RED-SVM use traditional features, while the others deep features. The performance gaps between these two approaches are not huge, since 1,050 images are insufficient for training deep networks.

## 5 IMPACTS OF APPLICATIONS

The proposed algorithm can be applied to various ranking problems. In this paper, we demonstrated three vision applications: facial age estimation, aesthetic score regression, and historical image classification. In particular, the proposed age estimator has various potential uses. For example, it can block or recommend media contents to people according to their ages. However, it has harmful impacts, as well as positive ones. Moreover, although age information lacks the distinctiveness to identify an individual, identity features, extracted by ORID, can be misused in facial recognition systems, causing serious problems such as unwanted invasion of privacy (Raji et al., 2020). Hence ethical considerations should be made before the use of the proposed algorithm.

Recently, ethical concerns about the fairness and safety of automated systems have been raised (Castelvecchi, 2020; Roussi, 2020; Noorden, 2020). Especially, due to the intrinsic imbalance of facial datasets (Ricanek & Tesafaye, 2006; Zhang et al., 2017; Niu et al., 2016), most deep learning methods on facial analysis (Wen et al., 2020; Or-El et al., 2020) have unwanted gender or racial bias. The proposed algorithm is not free from this bias either. Hence, before any practical usage, the bias should be resolved. Also, even though the proposed algorithm groups data in an *unsupervised* manner, data are clustered according to genders or races on MORPH II. These results should never be misinterpreted in such a way as to encourage any racial or gender discrimination. We recommend using the proposed age estimator for research only.

## 6 CONCLUSIONS

The DRC algorithm of ordered data based on ORID was proposed in this work. First, the ORID network decomposes the information of an object into the order and identity features. Then, DRC groups objects into clusters using their identity features in a repulsive manner. Also, we can estimate the rank of an unseen test by comparing it with references within the corresponding cluster based on the MAP decision. Extensive experimental results on various ordered data demonstrated that the proposed algorithm provides excellent clustering and rank estimation performances.

### ACKNOWLEDGMENTS

This work was supported in part by the MSIT, Korea, under the ITRC support program (IITP-2020-2016-0-00464) supervised by the IITP, and in part by the National Research Foundation of Korea (NRF) through the Korea Government (MSIP) under Grant NRF-2018R1A2B3003896.

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

## A  NETWORK STRUCTURE OF ORID

As described in Section 3.2, the ORID network consists of the encoder $F$, the decoder $G$, the comparator $C$, and the discriminator $D$. The network structures of these components are detailed in Tables 4$\sim$ 7, where '$k_h \times k_w$-$s$-$c$ Conv' and '$k_h \times k_w$-$s$-$c$ Deconv' denote the 2D convolution and 2D deconvolution with kernel size $k_h \times k_w$, stride $s$, and $c$ output channels, respectively. 'BN' means batch normalization (Ioffe & Szegedy, 2015), and '$c$ Dense' is a dense layer with $c$ output channels. Note that the encoder takes a $64 \times 64$ RGB image as input, and the identity feature of the encoder output is $l_2$-normalized in Eq. (3). Also, we set $d_{\mathrm{or}} = 128$ and $d_{\mathrm{id}} = 896$.

Table 4: The encoder $F$ in the ORID network.

| Layers | Output |
|---|---|
| 4×4-2-64 Conv BN LeakyReLU | 32×32×64 |
| 4×4-1-64 Conv BN LeakyReLU | 32×32×64 |
| 3×3-2-128 Conv BN LeakyReLU | 16×16×128 |
| 3×3-1-128 Conv BN LeakyReLU | 16×16×128 |
| 3×3-2-256 Conv BN LeakyReLU | 8×8×256 |
| 3×3-1-256 Conv BN LeakyReLU | 8×8×256 |
| 3×3-2-512 Conv BN LeakyReLU | 4×4×512 |
| 3×3-1-512 Conv BN LeakyReLU | 4×4×512 |
| 3×3-2-512 Conv BN LeakyReLU | 2×2×512 |
| 3×3-1-512 Conv BN LeakyReLU | 2×2×512 |
| 2×2-2-1024 Conv BN LeakyReLU | 1×1×1024 |

Table 5: The decoder $G$ in the ORID network.

| Layers | Output |
|---|---|
| 2×2-2-1024 Deconv BN Dropout(0.3) LeakyReLU | 2×2×1024 |
| 3×3-2-512 Deconv BN Dropout(0.3) LeakyReLU | 4×4×512 |
| 4×4-2-512 Deconv BN LeakyReLU | 8×8×512 |
| 4×4-2-256 Deconv BN LeakyReLU | 16×16×256 |
| 4×4-2-128 Deconv BN LeakyReLU | 32×32×128 |
| 4×4-2-3 Deconv Tanh | 64×64×3 |

Table 6: The comparator $C$ in the ORID network.

| Layers | Output |
|---|---|
| 256 Dense BN LeakyReLU | 256 |
| 256 Dense BN LeakyReLU | 256 |
| Dropout(0.5) | 256 |
| 3 Dense Softmax | 3 |

Table 7: The discriminator $D$ in the ORID network.

| Layers | Output |
|---|---|
| 5×5-2-64 Conv BN LeakyReLU | 32×32×64 |
| Dropout(0.5) | 32×32×64 |
| 5×5-2-128 Conv BN LeakyReLU | 16×16×128 |
| 5×5-2-128 Conv BN LeakyReLU | 8×8×128 |
| Dropout(0.3) | 8×8×128 |
| Reshape | 8192 |
| 1×1-1-1 Conv | 1 |

# B  ALGORITHMS – DETAILS

## B.1  OPTIMALITY OF CENTROID RULE

To solve the constrained optimization problem in Eq. (7), we construct the Lagrangian function

$$L = \sum_{j=1}^{k} \sum_{x \in \mathcal{C}_j} \left( (h_{\text{id}}^x)^t c_j - \alpha \frac{1}{k-1} \sum_{l \neq j} (h_{\text{id}}^x)^t c_l \right) - \lambda_j \sum_{j=1}^{k} (c_j{}^t c_j - 1) \tag{14}$$

where $\lambda_j, 1 \leq j \leq k$, are Lagrangian multipliers (Bertsekas, 1996). By differentiating $L$ with respect to $c_j$ and setting it to zero, we have

$$\frac{\partial L}{\partial c_j} = \sum_{x \in \mathcal{C}_j} h_{\text{id}}^x - \alpha \frac{1}{k-1} \sum_{l \neq j} \sum_{x \in \mathcal{C}_l} h_{\text{id}}^x - 2\lambda_j c_j \tag{15}$$

$$= \sum_{x \in \mathcal{C}_j} h_{\text{id}}^x - \alpha \frac{1}{k-1} \sum_{x \in \mathcal{X} \setminus \mathcal{C}_j} h_{\text{id}}^x - 2\lambda_j c_j \tag{16}$$

$$= 0 \tag{17}$$

for $j = 1, \ldots, k$. Therefore, the optimal centroid $c_j$ is given by

$$c_j = \frac{\sum_{x \in \mathcal{C}_j} h_{\text{id}}^x - \alpha \frac{1}{k-1} \sum_{x \in \mathcal{X} \setminus \mathcal{C}_j} h_{\text{id}}^x}{2\lambda_j}. \tag{18}$$

Because of the normalization constraint $c_j{}^t c_j = 1$, we have

$$2\lambda_j = \left\| \sum_{x \in \mathcal{C}_j} h_{\text{id}}^x - \alpha \frac{1}{k-1} \sum_{x \in \mathcal{X} \setminus \mathcal{C}_j} h_{\text{id}}^x \right\|, \tag{19}$$

which leads to the centroid rule in Eq. (8).

## B.2  OPTIMALITY OF NN RULE

Let us consider two cases. First, instance $x$ is declared to belong to cluster $\mathcal{C}_j$. It then contributes to the objective function $J$ in Eq. (6) by

$$\beta_j = (h_{\text{id}}^x)^t c_j - \alpha \frac{1}{k-1} \sum_{l \neq j} (h_{\text{id}}^x)^t c_l. \tag{20}$$

Second, $x$ is declared to belong to another cluster $\mathcal{C}_{j'}$. Then, its contribution is

$$\beta_{j'} = (h_{\text{id}}^x)^t c_{j'} - \alpha \frac{1}{k-1} \sum_{l \neq j'} (h_{\text{id}}^x)^t c_l. \tag{21}$$

By comparing the two contributions, we have

$$\beta_j - \beta_{j'} = (h_{\text{id}}^x)^t (c_j - c_{j'}) - \alpha \frac{1}{k-1} (h_{\text{id}}^x)^t (c_{j'} - c_j) \tag{22}$$

$$= \left(1 + \alpha \frac{1}{k-1}\right)(h_{\text{id}}^x)^t (c_j - c_{j'}). \tag{23}$$

This means that $\beta_j \geq \beta_{j'}$ when $(h_{\text{id}}^x)^t c_j \geq (h_{\text{id}}^x)^t c_{j'}$. Therefore, $x$ should be assigned to the optimal cluster $\mathcal{C}_{j*}$ such that the cosine similarity $(h_{\text{id}}^x)^t c_{j*}$ is maximized. Equivalently, we have the NN rule in Eq. (9).

## B.3  REGULARIZATION CONSTRAINT IN DRC

To prevent empty clusters and balance the partitioning, we enforce a regularization constraint so that every cluster contains at least a predefined number of instances. More specifically, when applying the NN rule, we enforce that at least $\frac{1}{2k}$ of instances are assigned to each cluster $\mathcal{C}_j$. The instances are selected in the decreasing order of cosine similarity $(h_{\text{id}}^x)^t c_j$.

## B.4  LOSS FUNCTIONS

In the DRC-ORID algorithm, we use the loss function

$$\ell = \lambda_{\text{rec}} \ell_{\text{rec}} + \lambda_{\text{clu}} \ell_{\text{clu}} + \lambda_{\text{com}} \ell_{\text{com}} + \lambda_{\text{gan}} \ell_{\text{gan}} \tag{24}$$

where the reconstruction, clustering, comparator, and GAN losses are given by

$$\ell_{\text{rec}} = \frac{1}{2N} \sum_{i=1}^{N} \left( \|x_i - G(F(x_i))\|_1 + \|y_i - G(F(y_i))\|_1 \right), \tag{25}$$

$$\ell_{\text{clu}} = -\frac{1}{2N} \sum_{i=1}^{N} \left( (h_{\text{id}}^{x_i})^t c_j + (h_{\text{id}}^{y_i})^t c_j \right), \tag{26}$$

$$\ell_{\text{com}} = -\frac{1}{N} \sum_{i=1}^{N} \left( q_{\succ}^{x_i y_i} \log p_{\succ}^{x_i y_i} + q_{\approx}^{x_i y_i} \log p_{\approx}^{x_i y_i} + q_{\prec}^{x_i y_i} \log p_{\prec}^{x_i y_i} \right), \tag{27}$$

$$\ell_{\text{gan}} = -\frac{1}{2N} \sum_{i=1}^{N} \left( \log(1 - D(G(F(x_i)))) + \log(1 - D(G(F(y_i)))) \right), \tag{28}$$

respectively. Here, $N$ is the number of image pairs in a minibatch. The weighting parameters are set to $\lambda_{\text{rec}} = 5$, $\lambda_{\text{clu}} = 0.1$, $\lambda_{\text{com}} = 1$, and $\lambda_{\text{gan}} = 1$.

## B.5 IMPACTS OF REPULSIVE TERM ON CLUSTERING

To analyze the impacts of the repulsive term in Eq. (6), we first compare clustering qualities with $\alpha = 0$ and $\alpha = 0.1$. At $\alpha = 0$, the repulsive term is excluded from the objective function $J$ and the centroid rule is reduced to Eq. (10) in the spherical $k$-means (Dhillon & Modha, 2001). However, different from the spherical $k$-means, even at $\alpha = 0$, the clustering is jointly performed with the training of the ORID network.

We adopt two metrics to measure the quality of clustering: normalized mutual information (NMI) (Strehl & Ghosh, 2002) and centroid affinity (CA). NMI measures the information shared between two different partitioning of the same data $A = \cup_{i=1}^{U} A_i$ and $B = \cup_{j=1}^{V} B_j$,

$$\text{NMI}(A, B) = \frac{\sum_{i=1}^{U} \sum_{j=1}^{V} |A_i \cap B_j| \log \frac{N|A_i \cap B_j|}{|A_i||B_j|}}{\sqrt{\left(\sum_{i=1}^{U} |A_i| \log \frac{|A_i|}{N}\right)\left(\sum_{j=1}^{V} |B_j| \log \frac{|B_j|}{N}\right)}} \tag{29}$$

where $U$ and $V$ are the numbers of clusters in $A$ and $B$, respectively, $N$ is the total number of samples, and $|\cdot|$ denotes the cardinality. Also, we define the centroid affinity (CA) as

$$\text{CA}(\{c\}_{j=1}^{k}) = \frac{2}{k(k-1)} \sum_{j=1}^{k} \sum_{l>j}^{k} c_j^t c_l. \tag{30}$$

For high-quality clustering, the centroids should be far from one another and thus should yield a low CA score.

Figure 10 plots how NMI and CA vary as the iteration goes on. In this test, MORPH II (setting B) is used and the number of clusters $k$ is set to 2. Since setting B consists of Africans and Caucasians, we use the race groups as the ground-truth partitioning for the NMI measurement. At early iterations, the NMI score of DRC-ORID with $\alpha = 0.1$ is slightly better than that with $\alpha = 0$. However, as the iterative training and clustering go on, the score gap gets larger. After the convergence, DRC-ORID with $\alpha = 0.1$ outperforms the option $\alpha = 0$ by a significant NMI gap of 0.13. Also, CA of the option $\alpha = 0.1$ gradually decreases, whereas that of $\alpha = 0$ does not. At $\alpha = 0.1$, the repulsive term makes the centroids repel each other. As a result, CA, which is the cosine similarity between the two centroids, becomes almost $-1$, which means the equilibrium state in Eq. (11) is almost achieved.

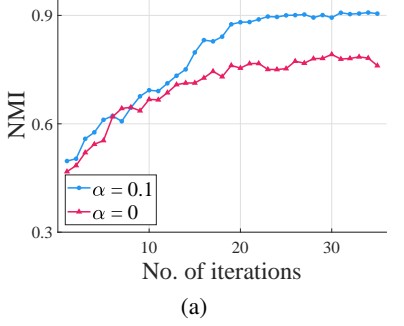

(a)

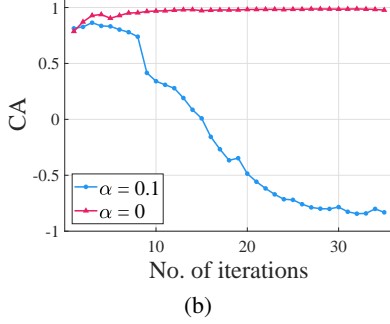

(b)

Figure 10: Comparison of (a) NMI and (b) CA curves with and without the repulsive term.

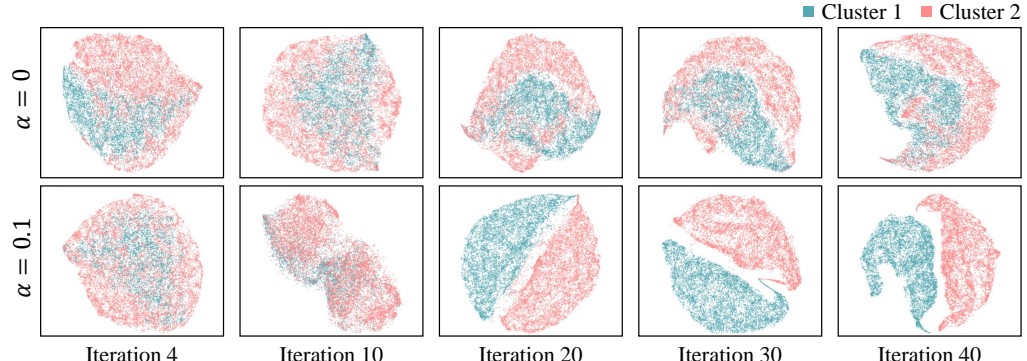

Figure 11: Comparison of the feature space transition of MORPH II (setting B) at $\alpha = 0$ and $\alpha = 0.1$.

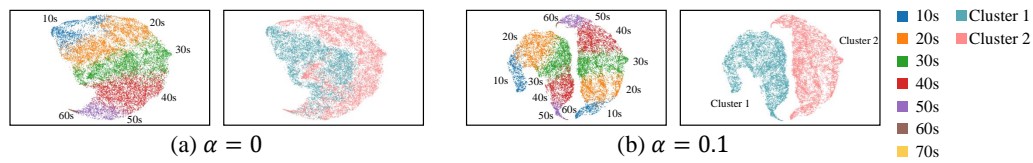

Figure 12: t-SNE visualization of the feature spaces of MORPH II (setting B) with age labels after the convergence.

We also visualize the feature spaces of the two options, $\alpha = 0$ and $\alpha = 0.1$, using t-SNE in Figure 11. It is observed that two clusters are more clearly separated by DRC-ORID with $\alpha = 0.1$. Figure 12 shows the t-SNE results after the convergence with age labels.

Figure 13 compares the NMI curves at different $\alpha$'s. The choice of $\alpha$ affects the quality of clustering, as $\alpha$ controls the intensity of the repulsive force between centroids. When $\alpha$ is too large, the centroids move too quickly, making the training of the ORID network difficult. On the other hand, when $\alpha$ is too small, the repulsive term does not affect the clustering meaningfully. Hence, $\alpha$ should be selected to strike a balance between training reliability and effective repulsion. It was found experimentally that clustering is performed well around $\alpha = 0.1$.

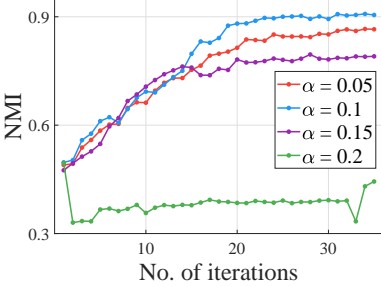

Figure 13: Comparison of NMI curves at different $\alpha$'s.

Finally, it is worth pointing out that, if the identity features were not normalized as in Eq. (3) and the repulsive clustering were performed in an unbounded space, the distances between centroids would get larger and larger as the iteration goes on. Thus, convergence would not be achieved. This is why we perform DRC on the bounded unit sphere.

### B.6 IMPACTS OF REPULSIVE TERM ON RANK ESTIMATION

Table 8 compares the rank estimation results when the clustering is performed with and without the repulsive term. In this experiment, we use MORPH II (setting A) and set $k = 2$. Without the repulsive term, lower-quality clusters make the training of the comparator more difficult. As a result, the age estimation performance degrades significantly in terms of both MAE and CS. In other words, the quality of clustering affects the rank estimation performance greatly, and the proposed DRC algorithm provides high quality clusters suitable for the rank estimation.

Table 8: Comparison of age estimation results on MORPH II (setting A) when clustering is performed with and without the repulsive term.

| Algorithm | MAE | CS (%) |
|---|---|---|
| Proposed without repulsive term | 2.47 | 90.7 |
| Proposed with repulsive term | 2.26 | 93.8 |

### B.7 IMPACTS OF THE NUMBER $k$ OF CLUSTERS ON RANK ESTIMATION

Tables 9 and 10 compare the rank estimation results according to the number $k$ of clusters on the MORPH II (setting A) and AADB datasets, respectively. On MORPH II, the age estimation performance decreases as $k$ increases. Since the training set in setting A consists of only 4,394 images, each cluster at a large $k$ contains too few instances. Thus, the comparator is trained inefficiently with fewer training pairs, degrading the performance. In contrast, AADB contains a large number of diverse images. Due to the diversity, a relatively large $k$ should be used to group images into meaningful clusters. Also, even at a large $k$, each cluster contains a sufficient number of data. Thus, as compared to MORPH II, results on AADB are less sensitive to $k$. In addition, we provide age estimation results on the balanced dataset in Table 14, in which $k$ has marginal impacts on the rank estimation performance.

As mentioned previously, the quality of clustering significantly affects the rank estimation performance. Also, similarly to other algorithms based on $k$-means, the clustering quality of DRC is affected by $k$. Hence, for the proposed algorithm to be used on a new ordered dataset, $k$ should be determined effectively to obtain good clustering and rank estimation results. Readers interested in the selection of $k$ are referred to Pham et al. (2005).

Table 9: Age estimation results according to $k$ on MORPH II (setting A).

| Algorithm | MAE | | | CS (%) | | |
|---|---|---|---|---|---|---|
| | $k = 2$ | $k = 3$ | $k = 4$ | $k = 2$ | $k = 3$ | $k = 4$ |
| Proposed | 2.26 | 2.32 | 2.43 | 93.8 | 92.7 | 91.4 |

Table 10: Aesthetic score regression results according to $k$ on AADB.

| Algorithm | $k = 4$ | $k = 6$ | $k = 8$ | $k = 10$ |
|---|---|---|---|---|
| Proposed | 0.1073 | 0.1059 | 0.1056 | 0.1060 |

### B.8 CLUSTERING USING OTHER FEATURES

Instead of clustering identity features $h_{\text{id}}^{x_1}, h_{\text{id}}^{x_2}, \ldots, h_{\text{id}}^{x_n}$, we test clustering order features $h_{\text{or}}^{x_1}, h_{\text{or}}^{x_2}, \ldots, h_{\text{or}}^{x_n}$ or whole features $h^{x_1}, h^{x_2}, \ldots, h^{x_n}$. In this test, MORPH II (setting A) is used and $k = 2$. Figure 14 compares the clustering results. When using order features or whole features, instances are divided by their ages. We see that instances younger than 30 mostly belong to cluster 1 and the others to cluster 2. Table 11 compares the performances of the age estimators trained using these clustering results. The best performance is achieved when the clustering is done on identity features.

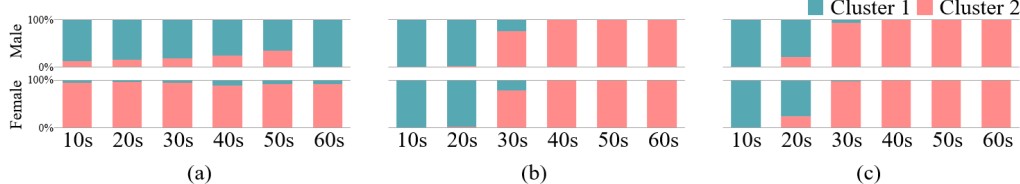

Figure 14: Clustering results using different features: (a) identity features, (b) order features, and (c) whole features.

Table 11: Age estimation performances when the three clustering results in Figure 14 are used.

| Feature type | MAE | CS (%) |
|---|---|---|
| Order | 2.42 | 91.5 |
| Whole | 2.59 | 90.7 |
| Identity (proposed) | 2.26 | 93.8 |

### B.9 RELIABILITY OF FEATURE DECOMPOSITION

Performing the comparison using order features only does not theoretically guarantee that order-related information is fully excluded from identity features. However, we observed empirically that the decomposition is sufficiently reliable if the dimension of an identity feature is selected properly. If the dimension is too small, the encoder may lose a significant portion of order-irrelevant information. On the contrary, if the dimension is too large, the encoder may encode order information redundantly. In our experiments, we use 128 and 896 dimensional vectors for order and identity features ($d_{or} = 128$ and $d_{id} = 896$), and obtain satisfactory decomposition results.

To show that order-related information is excluded from identity features, we compare the accuracies of the comparator (*i.e.* ternary classifier), when identity features are used instead of order features. Specifically, we first extract order features and identity features from all instances in MORPH II using the pretrained ORID network. Then, we train two comparators that predict the ordering relationship between two instances $x$ and $y$: one takes the order features $h_{or}^x$ and $h_{or}^y$ as input and the other takes the identity features $h_{id}^x$ and $h_{id}^y$. Table 12 lists the comparator accuracies. We see that the comparator fails to predict ordering relationships from identity features.

Also, Figure 15 is t-SNE visualization of the identity feature spaces with age or cluster labels, which confirms that order-related information is excluded effectively from identity features.

Table 12: Comparison of the comparator accuracies when different input features are used.

| | Order feature | Identity feature | Random guess |
|---|---|---|---|
| MORPH II (setting A) | 81.4 | 43.2 | 33.3 |
| MORPH II (setting B) | 75.4 | 40.8 | 33.3 |

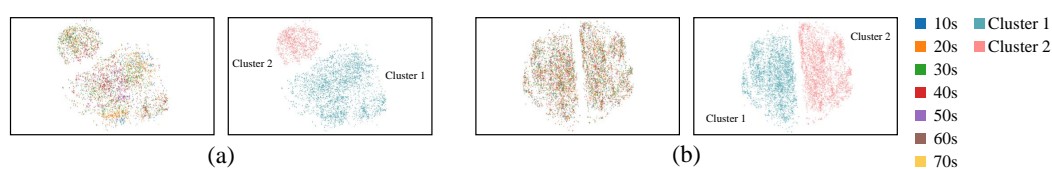

Figure 15: t-SNE visualization of identity feature spaces of MORPH II with age or cluster labels: (a) setting A and (b) setting B.

### B.10 MAP ESTIMATION

Let us describe the MAP estimation rule for rank estimation in Section 3.4. Given a test instance $x$, we select references $y_i$ by Eq. (13). Then, by comparing $x$ with $y_i$, the comparator yields the probability vector $p^{xy_i} = (p^{xy_i}_{\succ}, p^{xy_i}_{\approx}, p^{xy_i}_{\prec})$ for the three cases in Eq. (4). Thus, given $y_i$, the probability of $\theta(x) = r$ can be written as

$$P_{\theta(x)}(r \mid y_i) = p^{xy_i}_{\succ} \cdot P_{\theta(x)}(r \mid x \succ y_i) + p^{xy_i}_{\approx} \cdot P_{\theta(x)}(r \mid x \approx y_i) + p^{xy_i}_{\prec} \cdot P_{\theta(x)}(r \mid x \prec y_i). \quad (31)$$

Suppose that $x \succ y_i$. Then, $\theta(x) - \theta(y_i) = r - i > \tau$ from Eq. (4). Also, the maximum possible rank is $m$. We hence assume that $\theta(x)$ has the uniform distribution between $i + \tau + 1$ and $m$. In other words, $P_{\theta(x)}(r \mid x \succ y_i) \sim U(i + \tau + 1, m)$, where $U$ denotes a discrete uniform distribution. Similarly, we have $P_{\theta(x)}(r \mid x \approx y_i) \sim U(i - \tau, i + \tau)$ and $P_{\theta(x)}(r \mid x \prec y_i) \sim U(1, i - \tau - 1)$. Then, we approximate the *a posteriori* probability $P_{\theta(x)}(r \mid y_1, \ldots y_m)$ by averaging those single-reference inferences in Eq. (31);

$$P_{\theta(x)}(r \mid y_1, \ldots y_m) = \frac{1}{m} \sum_{i=1}^{m} P_{\theta(x)}(r \mid y_i). \quad (32)$$

Finally, we obtain the MAP estimate of the rank of $x$, which is given by

$$\hat{\theta}(x) = \arg\max_{r \in \Theta^l} P_{\theta(x)}(r \mid y_1, \ldots y_m). \quad (33)$$

## C  FACIAL AGE ESTIMATION – MORE EXPERIMENTS AND DETAILS

### C.1  IMPLEMENTATION DETAILS

We initialize the parameters of the ORID network for facial age estimation using the Glorot normal method (Glorot & Bengio, 2010). We use the Adam optimizer with a learning rate of $10^{-4}$ and decrease the rate by a factor of 0.5 every 50,000 steps. For data augmentation, we do random horizontal flips only. This is because other augmentation schemes, such as brightness or contrast modification, may deform identity information such as skin colors. Also, $d_{\text{or}}$ and $d_{\text{id}}$ are set to be 128 and 896, respectively. In Eq. (6), we set $\alpha$ to 0.1 and decrease it to 0.05 after 200 epochs.

### C.2  EVALUATION SETTINGS

For evaluation on the MORPH II dataset, we adopt four widely used testing protocols.

- Setting A – 5,492 images of the Caucasian race are selected and then randomly divided into two non-overlapping parts: 80% for training and 20% for test.
- Setting B – 21,000 images of Africans and Caucasians are selected to satisfy two constraints: the ratio between Africans and Caucasians should be 1 : 1, and that between females and males 1 : 3. They are split into three disjoint subsets S1, S2, and S3. The training and testing are repeated twice: 1) training on S1, testing on S2 + S3, and 2) training on S2, testing on S1 + S3. The average performance of the two experiments is reported.
- Setting C – This setting is the 5-fold cross-validation on the entire dataset. Images are randomly split into five folds, but the same person's images should belong to only one fold. The average performance of the five experiments is reported.
- Setting D – This is called the 80-20 protocol. Without any constraint, the entire dataset is randomly divided into the training and test sets with ratio 8 : 2. Thus, setting D is similar to one experiment in setting C, but the same person's images may belong to both training and test sets.

### C.3  CLUSTERING

We provide more clustering results on MORPH II. Figure 16 is the clustering results on setting B at $k = 2$. Since setting B consists of Africans and Caucasians, the images are clustered according to the races. Also, Table 13 summarizes the clustering results for settings A, B, and C at $k = 2$. The clustering result on setting D is omitted, since it is almost identical with that on setting C. In all settings, the proposed DRC-ORID divides facial images into two clusters with meaningful criteria, which are gender for setting A and race for settings B, C, and D.

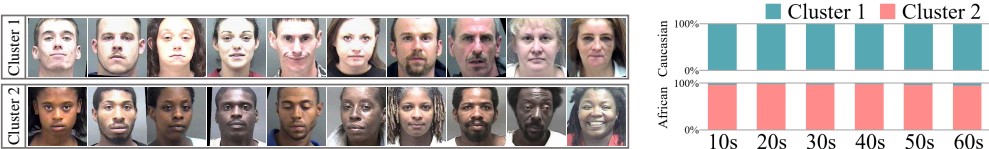

Figure 16: MORPH II images in setting B are divided into two clusters.

Table 13: Clustering results on settings A, B, and C of MORPH II at $k = 2$.

| Cluster | Setting A | | Setting B | | Setting C | | | | |
|---|---|---|---|---|---|---|---|---|---|
| | Male | Female | Caucasian | African | Caucasian | African | Hispanic | Asian | Others |
| 1 | 897 | 196 | 3,507 | 63 | 8 | 33,008 | 4 | 0 | 46 |
| 2 | 209 | 3,152 | 26 | 3,404 | 8,427 | 1,065 | 115 | 50 | 1,384 |

### C.4  AGE ESTIMATION

We implement a VGG-based pairwise comparator and follow the settings of Lim et al. (2020). Specifically, instead of Eq. (4), we use the ternary categorization based on the geometric ratio and

set $\tau = 0.1$. We initialize its feature extractor using VGG16 pre-trained on the ILSVRC2012 dataset (Deng et al., 2009) and its fully connected layers using the Glorot normal method. We employ the Adam optimizer with a minibatch size of 32. We start with a learning rate of $10^{-4}$ and shrink it by a factor of 0.5 after every 80,000 steps.

Table 14: Comparison of age estimation results on the balanced dataset.

| Algorithm | MAE | | | | CS (%) | | | |
|---|---|---|---|---|---|---|---|---|
| | $k=1$ | $k=2$ | $k=3$ | $k=6$ | $k=1$ | $k=2$ | $k=3$ | $k=6$ |
| MV (Pan et al., 2018) | 4.49 | 4.52 | 4.44 | 4.40 | 69.9 | 70.1 | 70.3 | 69.6 |
| OL-supervised (Lim et al., 2020) | 4.23 | 4.18 | 4.19 | 4.18 | 73.2 | 73.4 | 73.4 | 74.0 |
| OL-unsupervised (Lim et al., 2020) | - | 4.16 | 4.17 | 4.16 | - | 74.0 | 73.9 | 74.0 |
| Proposed | **4.15** | **4.01** | **4.02** | **4.05** | **74.0** | **74.8** | **74.8** | **74.6** |

Table 14 lists age estimation results on the balanced dataset according to the number $k$ of clusters. OL-supervised trains the comparator using supervised clusters separated according to gender or ethnic group annotations. Specifically, the supervised clusters at $k = 2, 3$, and $6$ are divided according to genders, ethnic groups, and both genders and ethnic groups, respectively. On the other hand, OL-unsupervised and the proposed algorithm determine their clusters in unsupervised manners. We see that the proposed algorithm performs better than the conventional algorithms in all tests. By employing multiple clusters, the proposed algorithm improves MAE by 0.12 and CS by 0.73% on average. In contrast, OL-unsupervised improves MAE by 0.04 and CS by 0.07% only. This indicates that, by employing identity features, the proposed DRC-ORID algorithm groups instances into meaningful clusters, in which instance ranks can be compared more accurately.

## C.5 Age Transformation

More age transformation results are in Figure 17. Note that, in Figure 6, given an image $x$, we select the reference $y$ at a target age, whose identity feature is the most similar to that of $x$, as in Eq. (13). Hence, the image $x$ and the reference $y$ have similar appearance. On the other hand, Figure 17 shows transformed images using randomly selected references. The first two cases transform the same image $x$ with different references, but the transformed images are similar. Also, even when the gender and/or race of $y$ are different from those of $x$, the identity information of $x$ is preserved well in the transformed image. This confirms the reliability of ORID.

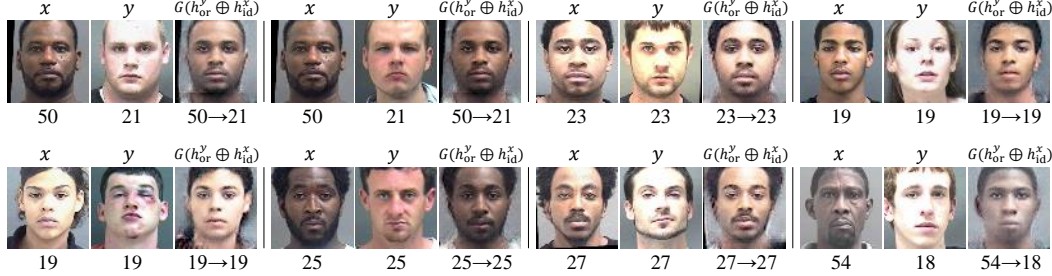

Figure 17: More age transformation results.

## C.6 Reconstruction

Figure 18 shows reconstructed faces using whole feature ($h_{\text{or}}^x \oplus h_{\text{id}}^x$), order feature only ($h_{\text{or}}^x \oplus \mathbf{0}$), and identity feature only ($\mathbf{0} \oplus h_{\text{id}}^x$). Without the order feature, each decoded face is degraded but the person can be identified. In contrast, without the identity feature, the reconstruction is not related to the person except that it seems to be an average face of people at the same age as the person. These results confirm that order and identity features are complementary.

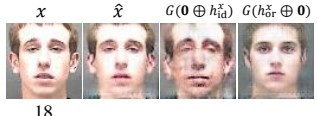 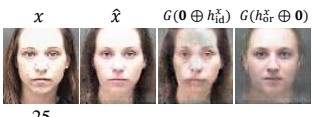 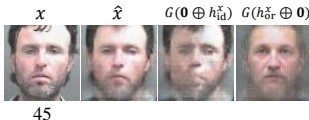

Figure 18: Reconstruction results. For each test, the input $x$, reconstruction $\hat{x} = G \circ F(x)$ using the whole feature, reconstruction $G(\mathbf{0} \oplus h_{\mathrm{id}}^x)$ using the identity feature, and reconstruction $G(h_{\mathrm{or}}^x \oplus \mathbf{0})$ using the order feature are shown.

## C.7 Reference Images

Figure 19 shows examples of reference images, which are used for the rank estimation on MORPH II (setting D) at $k = 2$. Given a test image $x$, reference image $y_i$ of age class $i$ is selected via Eq. (13) from the training set. In the default mode, for each age $i$, a single reference image is selected. However, the top $r$ most similar references can be selected and used for the estimation. We use a single reference because multiple references improve the estimation performance only negligibly. In Figure 19, the top three reference images are shown for each age from 16 to 53. In setting D, the two clusters are divided by Africans and the others in general. However, we see that test and reference images tend to have the same gender, as well as the same race. Furthermore, they have similar appearance, even when they have a big age difference.

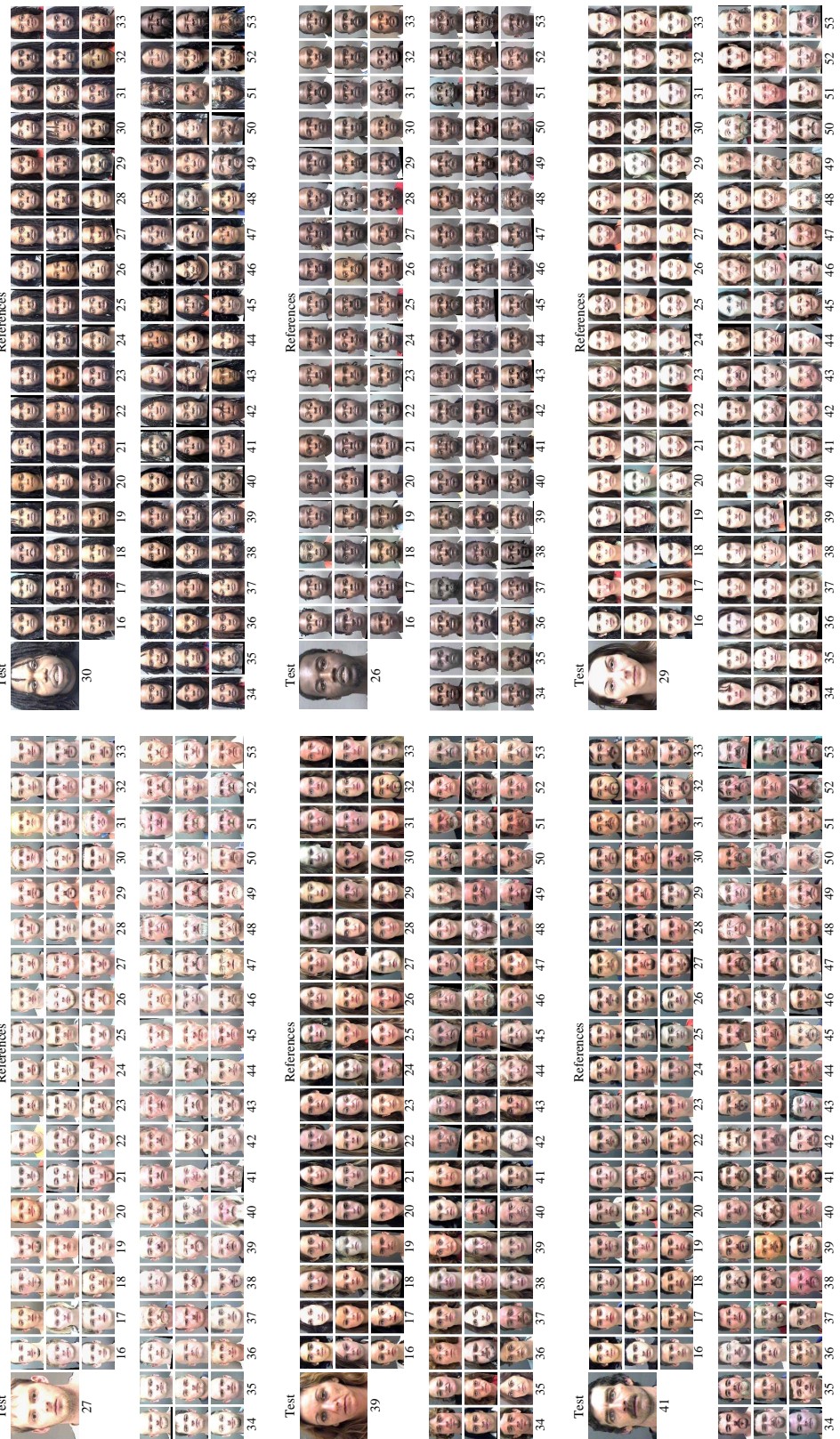

Figure 19: Examples of reference images in facial age estimation.

# D   AESTHETIC SCORE REGRESSION – MORE EXPERIMENTS AND DETAILS

## D.1   IMPLEMENTATION DETAILS

For aesthetic score regression, we implement a pairwise comparator based on EfficientNetB4 (Tan & Le, 2019). The pairwise comparator has the same architecture as that for facial age estimation, except for the backbone network. To initialize the backbone, we adopt the parameters pre-trained on the ILSVRC2012 dataset. We initialize the other layers using the Glorot normal method. We update the network parameters using the Adam optimizer with a minibatch size of 16. We start with a learning rate of $10^{-4}$ and shrink it by a factor of 0.8 every 8000 steps. Training images are augmented by random horizontal flipping. We set $\tau = 0.15$ for the ternary categorization in Eq. (4).

## D.2   CLUSTERING

Notice that the AADB dataset contains images of diverse contents and styles. Hence, when clustering with a small $k$, it is hard to observe the characteristics shared by images within each cluster, whereas $k = 2$ or 3 is sufficient for facial age data. We empirically found that at least eight clusters are required ($k = 8$) to partition the AADB dataset by meaningful criteria. Figure 20 provides more examples of clustering results at $k = 8$.

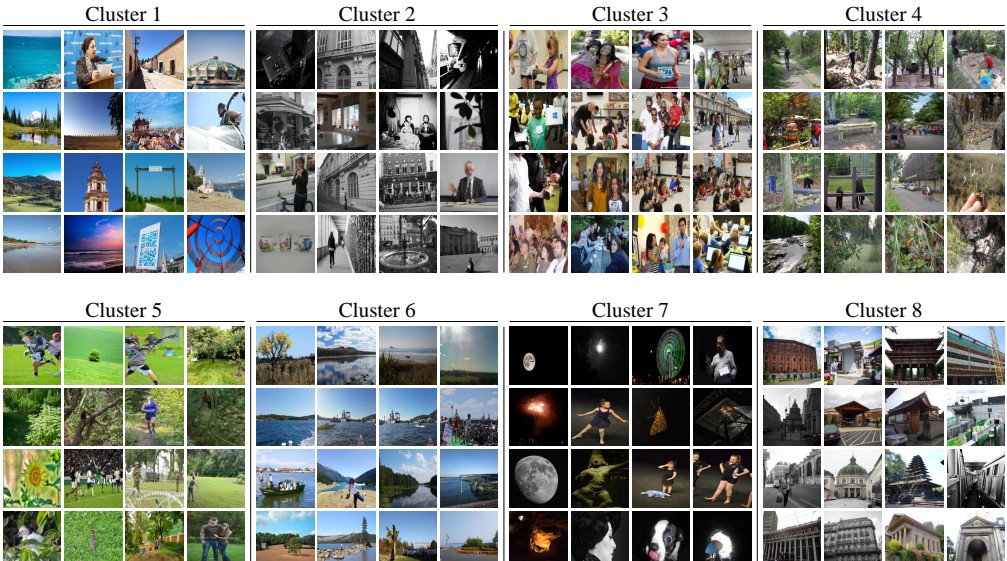

Figure 20: Example AADB images grouped into eight clusters ($k = 8$).

## D.3 REFERENCE IMAGES

Figure 21 shows examples of reference images, which are used for the aesthetic score regression. Given a test image $x$, the reference image $y_i$ of aesthetic class $i$ is selected by Eq. (13). For the aesthetic score regression, we use a single reference image for each aesthetic class, as done in the facial age estimation. Thus, 101 reference images are used in total.

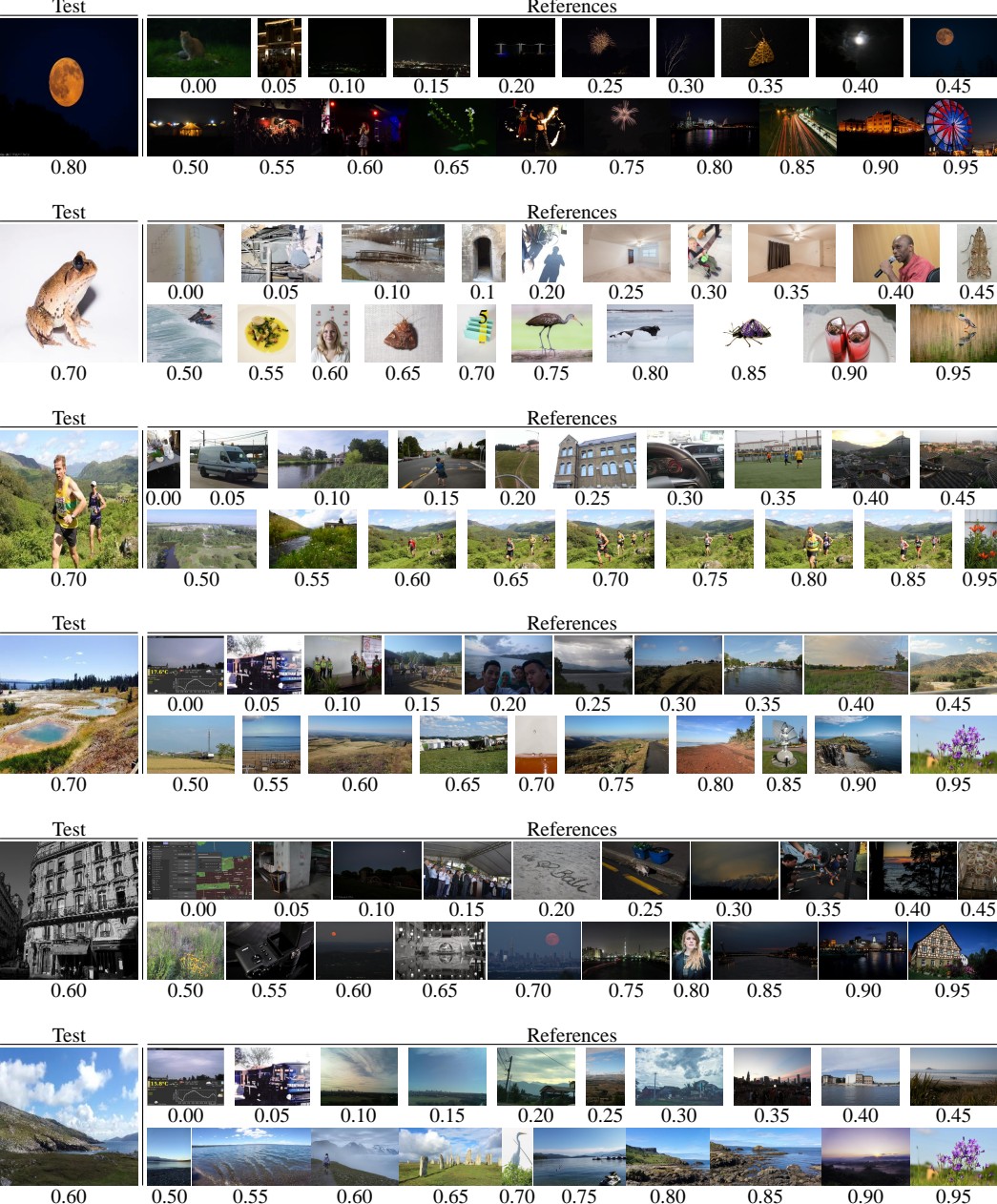

Figure 21: Examples of reference images in aesthetic score regression.

# E HCI CLASSIFICATION – MORE EXPERIMENTS AND DETAILS

## E.1 IMPLEMENTATION DETAILS

For DRC-ORID for HCI classification, we set all hyper-parameters in the same way as we do in Appendix C.1. We set $\tau = 1$ for the ternary categorization of ordering relationship in Eq. (4). Note that there are five decade classes from 1 to 5.

## E.2 CLUSTERING

Figure 22 shows some sample images in the HCI dataset, which are ordered according to the decade classes. Figure 23 shows more example HCI images grouped into four clusters ($k = 4$).

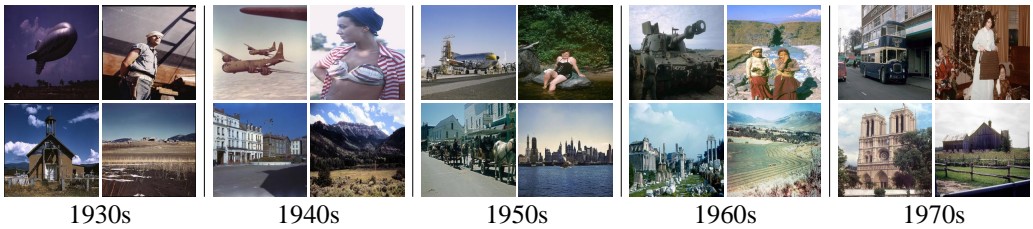

Figure 22: Example images in the HCI dataset.

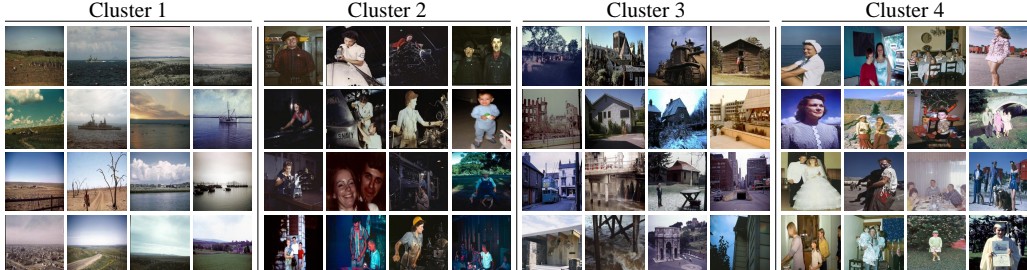

Figure 23: Example HCI images grouped into four clusters ($k = 4$).

## E.3 REFERENCE IMAGES

Figure 24 shows the five reference images for each of six test image examples. Note that, given a test image, the reference images of similar contents, tones, or composition are selected from the five decade classes.

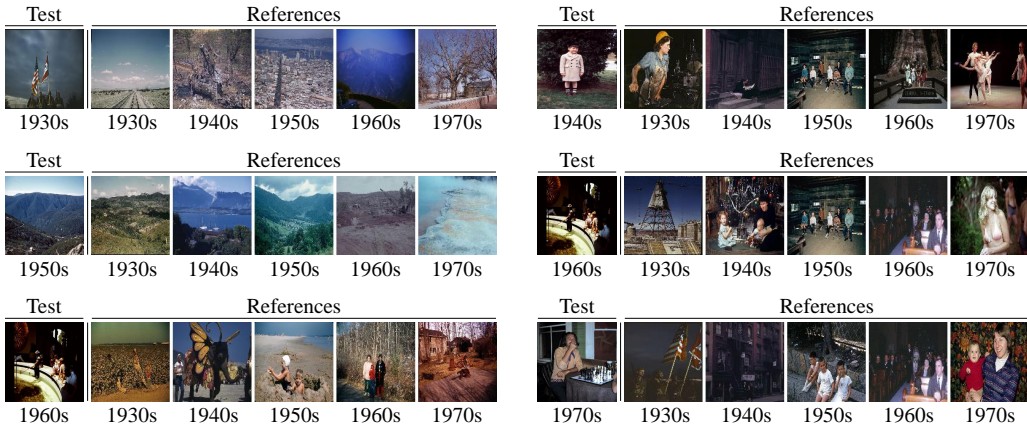

Figure 24: Examples of reference images in historical color image classification.

