# OpenReview forum: "Deep Repulsive Clustering of Ordered Data Based on Order-Identity Decomposition"
_ICLR.cc/2021/Conference — ICLR 2021 Poster_

### Official Review · AnonReviewer4 · 2020-10-15
**Official Blind Review #4**

**Rating:** 7
**Confidence:** 4

**Review:**

Summary:
This paper considers the problem of order learning, which learns an ordinal classification function. This paper proposes to learn separarted order-relavent and order-irrelavent latent representations to improve the performance of existing methods, which is a very interesting and promising idea. However, the approach lacks novelty and convincing theoretical guarantees, as well as not showing convincing performance even through the insufficient empirical evaluation.

Main concerns:
- The ORID model structure: The latent representation is separated to h_{or} and h_{id}, and the comparison loss is defined on h_{or}. However, this need not to exclude order-relavent information from h_{id}. Also, it needs to be clarified that to what exent introudcing a discriminator helps, as this turns a minimization problem into an unstable min-max optimization problems. How it works without the discriminator?

- Normalization of h_{id}: Normalizing vectors in a space may result a totally different cluster structure, different clusters may appear to be overlapped with each other by normalization. Euclidean distance can be the natural dissimilarity metric without normalization.

- The DRC algorithm: The idea of encouraging inner-cluster similarity and iter-cluster dissimilarity of Eq. (9) is not new. Also, right after Algorithm 1 in the paper, "DRC is guaranteed to converge to a local maximum" is quite suspicious. Is it true that different rules optimiziming the same objective alternatively is guaranteed to converge? At least some references need to be provided as it is a crucial point of the main contribution.

- The decisioin rule: Eq.(15) loops over all y, so what is the point of selecting a y_i in Eq.(13)?

- Experimental results seem to be fine, and authors are honest to report unfavorable results. However, in my humble opinion, results for a sufficient number of repetitions (5 or 10) is needed to achieve a least convincibility.

Minor comments:
- In Eq.(4), the rightmost inequation should be \theta(x) - \theta(y) < -r.

---

> ### Author Response · Authors · 2020-11-18
> **Response to reviewer 4**
>
> Thank you for your time and effort to review our manuscript. Please find our responses below.
> ***
> * **Decomposition reliability:** As you pointed out, performing comparisons using order features only does not guarantee theoretically that order-related information is fully excluded from identity features. However, we observed empirically that the decomposition is reliable enough. To demonstrate this, we showed age transformation results in Figures 6 and 17. Moreover, in Appendix B.9, we have tested the accuracy of the comparator (ternary classifier), when identity features are used instead of order features. The comparator fails to predict ordering relation using identity features, which indicates that order-related information is effectively excluded from identity features. Additional results supporting the reliability of ORID are available also in Appendix B.9 in p. 19.
>
> * **Use of discriminator:** We adopt the discriminator to obtain more realistic reconstructions. Without the discriminator, the ORID network is trained more stably but the decoder reconstructs more blurred faces. Except for blurry artifacts, clustering and rank estimation results are not affected noticeably by the removal of the discriminator. In fact, age estimation results are not changed at all. On the other hand, we should solve a relatively unstable minimax problem when using the discriminator. However, in our experiments, it needs only a few trials to find a proper set of parameters. This is because the goal of the decoder is just to reconstruct input images, which is a relatively easy task in the current deep learning technology. To avoid the repetition of parameter searching by other researchers, we have described the parameters in Appendix A in p. 14. Also, we will make training, as well as test, codes available.
>
> * **Normalization of $h_{\rm id}$:** DRC performs clustering on the $d_{\rm id}$-dimensional unit sphere. In other words, we cluster identity feature vectors $h_{\rm id}$ that are already $l_2$-normalized. Hence, the deformation of clustering structure through normalization does not occur. This has been clarified in 1st paragraph in p. 5.
> * **Novelty of DRC:** We agree that the idea of encouraging intra-cluster similarity and inter-cluster dissimilarity is not new. The same idea is the basis of a classic clustering paper [1]. However, to the best of our knowledge, DRC is the first attempt using an explicit repulsive term in deep clustering, which jointly optimizes clustering and feature embedding. This novelty has been clarified before Eq. (6) in p. 5.
>
>  [1] J. H. Ward, Jr. Hierarchical grouping to optimize an objective function. J. Am. Stat. Assoc., 1963.
>
> * **Convergence of DRC:** Yes, increasing the same function alternately using different rules guarantees convergence if the function is bounded above. This is because a monotonically increasing sequence, which is bounded above, is guaranteed to converge. In the revision, we have specified the bounding inequality, given by $J \leq  n + \frac{\alpha}{k-1}n$. Also, we have cited references [2], [3] as well for readers interested in the convergence. Please see the middle paragraph in p. 5.
>
>  [2] D. Pollard. Quantization and the method of k-means. IEEE Trans. Inf. Theory., 1982.
>
>  [3] M. J. Sabin and R. M. Gray. Global convergence and empirical consistency of the generalized Lloyd algorithm. IEEE Trans. Inf. Theory., 1986.
>
> * **Selection of $y_i$ in Eq. (13):** In Eq. (13), $y_i$ is selected among ${\mathcal C}_l$, whose identity feature is the most similar to input $x$. For example, in age estimation, $y_i$ is the reference face at age $i$, which looks the most similar to $x$. On the other hand, in Eq. (15) (Eq. (32) in the revision), $x$ is compared with reference images $y_i$ of all ages $i = 1, 2, \ldots, m$. Instead of selecting a single reference at each $i$, we can use multiple ones. However, this alternative does not increase the estimation performance meaningfully but needs more computations for more comparisons. This has been clarified in Appendix C.6 in p. 22.
>
> * **Multiple repetitions of experiments:** The rank estimation results of 5 repetitions on MORPH II (settings A and B) are shown below. The networks were trained from scratch 5 times. Note that the deviations are negligible, and even the worst repetition outperforms the conventional methods in Table 1. We are doing the repetitions for all experiments and will provide the results in the camera-ready.
>
>     | |\_\_\_\_\_\_MAE\_\_\_\_\_/\_\_\_\_CS (%)\_\_\_\_|
>     |-----------|:------------------:|
>     | Setting A | $2.264 \pm 0.01$ / $93.81 \pm 0.01$|
>     | Setting B | $2.516 \pm 0.02$ / $89.72 \pm 0.01$ |
>
> * **Error in Eq. (4):** It has been corrected. Thank you.
> ***
> We have revised our paper to address all your comments and hope that this revision resolve your concerns. If you have any additional concerns, please let us know.
>
> Thank you again for your constructive comments. We do appreciate them.

---

> > ### Comment · AnonReviewer4 · 2020-11-20
> > **Authors comment addressed most of my concerns**
> >
> > I really appreciate authors diligent work on resolving my concerns as well as those of other reviewers and improving the paper. I have raised my rating.

---

> > > ### Author Response · Authors · 2020-11-21
> > > **Thank you for your positive feedback**
> > >
> > > Thank you for your positive feedback on our responses. We do appreciate it.

---

### Official Review · AnonReviewer1 · 2020-10-27
**Missing discussion of broader impacts of application for an intuitive method for ordered data, also missing technical novelty / analysis**

**Rating:** 6
**Confidence:** 3

**Review:**

**Summary of paper**:

This paper considers the task ordered learning, making predicting a class label for a point among an ordered graph of classes. The paper proposes a clustering objective that encourages the model to separate data into groups such that classification prediction is easier within each cluster. The method is intuitive, clearly explained and well motivated. The paper indicates state of the art results on a task of estimating ages of individuals from photographs.

**Review summary**: Missing *crucial* discussion on discussion of use cases / broader impact of task of estimating ages from photographs. Otherwise intuitive and effective method for ordered data; effective empirical results; limited novelty / exploration of methodological approach.

**Strengths**:

The authors describe an intuitive and effective method for making predictions on ordered data. The approach uses a intuitive clustering-based method that groups data into subsets where items are easier to order. The paper is clearly written and explains the approach clearly. The paper shows several examples of predicted output of the method and shows results on two tasks (estimating ages, aesthetic score regression). The method achieves state of the art results on the task of estimating ages and is competitive on the other task. The authors show further results on age transformation.

**Weakness**:

**Broader Impacts of Applications**:  One of the primary applications of the paper is estimating ages of individuals based on their photographs. While this is paper is not the first to focus on such a task, it is very remiss of this paper to not discuss the motivations for this task and the broader impacts and ethical considerations of this task. I would very strongly encourage the authors to add a discussion of the potential uses of their system and the benefits (as well as harms) that come from these uses. I think that it is crucially important to discuss this both in the context of this work as well as previous work on the task. In particular, it would be important to mention how the use of clustering (into groups based on gender/race) in this model factors into potential biases when the model is used. I think it would be necessary to include this discussion in the body of the paper itself rather than an appendix. I greatly believe that this discussion is necessary and the lack of it is one of my top concerns about the paper.

**Distinctions between total ordering and partial ordered / related work**: The presentation of the approach indicates that observations are not directly comparable across clusters. However, the overall model does in fact provide a total ordering -- each point is mapped one of the clusters and then compared within that cluster. I think the presentation would be greatly improved if it were described not in a way that implies a partial ordering (only within each cluster) is there, but instead that the total ordering function is this multi-modal, cluster-based ordering. Further, I think it would important to discuss the relationships between this work and work on partially ordering sets, particularly work on combining partially ordered sets.  It might also be good to consider more related work on ordering, such as, Learning to Order Things  (https://papers.nips.cc/paper/1431-learning-to-order-things.pdf). Also, I think that it is especially important to address other work (such as that in extreme classification) that organizes class labels into groups that are easier to discriminate between (i.e.,  Logarithmic Time One-Against-Some ( https://arxiv.org/abs/1606.04988)).

**Novelty of approach / depth of exploration**: The core novelty of the approach is in the use of clustering to separate the data into groups that are easier to rank. This is a nice idea and appears give strong empirical benefits. I worry that since the clustering component is the core contribution of the paper, that the analysis of the method of clustering is not very deeply explored empirically. The idea is intuitive, but I feel the limited deviation from classic approaches that combine clustering + classification would benefit from additional analysis of the approach, along the dimension of the clustering objective that is selected.

**Questions for the authors:**

* What are the potential use cases for the system & its applications to age prediction?
* What are the fairness/ethical/safety concerns of such an application?
* Were clustering objectives other than the repulsive-based one considered?
* How does your work connect to papers such as Logarithmic Time One-Against-Some ( https://arxiv.org/abs/1606.04988) which also organize classes into clusters ?

---

> ### Author Response · Authors · 2020-11-18
> **Response to reviewer 1**
>
> Thank you for valuable comments and insightful suggestions. Please find our responses below.
> ***
> * **Broader impacts of applications:** We agree with you that it is important to discuss broader impacts of applications. We have devoted a new section in the main paper to the discussion of potential use cases and ethical concerns. Please see Section 5 in p. 9.
>
> * **Presentation without implying partial ordering:** As you suggested, instead of the partial ordering framework, we have presented the algorithm in the total ordering framework. Specifically, instead of assuming incomparability across chains, we assume that there is a total order in which each class represents $k$ subclasses of different types and that object instances of the same type are more easily compared. Please see 2nd paragraph in Section 3.1 in p. 3. We have revised the entire manuscript to remove the expressions implying partial ordering and describe them in the new framework. Thanks to your suggestion, the presentation has been improved greatly.
>
> * **Related work on combining partially ordered sets / Relation to "Learning to Order Things":** We have described how the proposed algorithm is related to the Cohen et al.'s "Learning to Order Things" algorithm and rank aggregation methods for combining partially ordered sets. Please see 2nd and 3rd paragraphs in Section 2.1 in p. 2.
>
> * **Connection to "Logarithmic Time One-Against-Some":** Similarly to the proposed algorithm, there are conventional approaches to use clustering ideas to perform classification or rank estimation more effectively, including "Logarithmic Time One-Against-Some." Whereas these approaches group data in the label dimension, the proposed algorithm cluster data in the dimension orthogonal to the label dimension. We have discussed how the proposed algorithm is related to and different from these approaches. Please see 2nd paragraph in p. 3.
>
> * **Empirical analysis of the clustering method and objective:** Impacts of the repulsive term was analyzed in detail in Appendix B.5 in p. 16. By comparing the two cases $\alpha = 0.1$ and $\alpha=0$, it was shown that the proposed DRC is more effective than the spherical $k$-means. Furthermore, we have added more experimental results on DRC in Appendices B.6 and B.7 in p. 18. In B.6, it has been shown that DRC helps to improve the rank estimation performance as well. In B.7, we have assessed age estimation performances when the clustering is performed using order features or whole features, instead of identity features. It has been shown that using identity features provides more accurate estimation results.
> ***
> We have made every attempt to address your comments in the revised manuscript and hope that you find this revision satisfactory. If you have additional concerns, please let us know.
>
> Thank you again for your constructive comments. We do appreciate them.

---

> ### Author Response · Authors · 2020-11-24
> **Section for impacts of applications has been strengthened**
>
> In the 2nd revision, we have strengthened **Section 5. Impacts of
> Applications** in p.9.
>
> Specifically, we have cited more references, including
> recent relevant articles in Nature [1-3], and have deepened the
> discussion on potential impacts of facial recognition systems in
> general and how the proposed algorithm should be employed in
> particular.
>
> Thank you again for your constructive comments.
>
>  ***
> [1] Davide Castelvecchi. Is facial recognition too biased to be let
> loose? Nature, 587(7834):347–349, 2020.
>
> [2] Richard Van Noorden. The ethical questions that haunt
> facial-recognition research. Nature, 587(7834):354–358, 2020.
>
> [3] Antoaneta Roussi. Resisting the rise of facial recognition.
> Nature, 587(7834):350–353, 2020.

---

### Official Review · AnonReviewer2 · 2020-10-29
**Well presented paper, some details need to be clarified.**

**Rating:** 6
**Confidence:** 3

**Review:**

- It is well presented. The idea of splitting the encoding feature space into task related features and non-task related features is probably not new. But the use of it in estimating rank might be new and intuitively it makes sense to use it. They also propose an extension to the clustering algorithm using a repulsive term and propose MAP estimation algorithm to assign a rank based on the output probabilities of the comparator when the max possible rank is known.
- Experiments are conducted on 3 data sets. The results show the effectiveness of the approach. The experiments, I feel, are sufficient to show that clustering instances based on non-rank related features will help improve effectiveness of comparison based ranking of new instances. They also show the effectiveness of their proposed MAP estimation rule for assigning a rank.
- The effectiveness of the repulsive clustering on ranking performance is not clear. The authors discuss that using the repulsive term in the objective for clustering produces more distinct clusters but how does this "improved" cluster quality translate to better performance in ranking? As this is one of the key contributions of the paper, a comparison of ranking performances with and without the use of the repulsive term in clustering would be useful.
- How sensitive/robust is the proposed approach to the number of clusters chosen? How can one choose the right number of clusters to use? A discussion on these would be useful.
- In each experiment, what was the dimensions of the order-related feature and identity-related feature?
In general, I think this paper is above the borderline. But I  would also like to see the comments from other reviewers.

---

> ### Author Response · Authors · 2020-11-18
> **Response to reviewer 2**
>
> Thank you for your positive review and constructive comments. Please check our responses below.
> ***
> * **Impacts of repulsive term on ranking performance:** The table below compares age estimation results when the clustering is performed with and without the repulsive term. In this test, we use MORPH II (setting A) and set $k=2$. Without the repulsive term, the clustering quality degrades as discussed in Appendix B.5. Such lower-quality clusters make the training of the comparator more difficult. Consequently, the age estimation performance is also lowered when the repulsive term is not used. We have discussed these impacts of the repulsive term on the ranking performance in Appendix B.6 in p.18.
>
>     |                        |\_\_\_MAE\_\_\_ |\_\_\_CS (%)\_\_\_ |
> |------------------------|:----:|:------:|
> | Without repulsive term | 2.47 | 90.7   |
> | With repulsive term    | 2.26 | 93.8   |
>
> * **Discussion on the number $k$ of clusters:**  As you suggested, we compare rank estimation results according to the number $k$ of clusters on the MORPH II and AADB datasets below. On MOPRH II (setting A), the age estimation performance (MAE/CS) decreases as $k$ increases. Since the training set of MORPH II (setting A) consists of 4,394 images only, each cluster at a large $k$ contains only a small number of instances. Thus, the comparator is trained less effectively with fewer training pairs, resulting in the performance drop. On the other hand, AADB contains a large number of diverse images. Due to its diversity, a relatively large $k$ facilitates grouping images into meaningful clusters. Also, even at a large $k$, each cluster contains a sufficiently large number of data.
>    \begin{array}{c|c|c|c}
> \hline  & k=2 & k=3 & k=4 \\\
> \hline
> \text{MORPH  II  (setting A)}  & 2.26/93.8 & 2.32/92.7  &2.43/91.4\\\
> \hline
> \end{array}
>    \begin{array}{c|c|c|c|c}
> \hline  & k=4 & k=6 & k=8 & k=10 \\\
> \hline
> \text{AADB}  & 0.1073 & 0.1059&0.1056&0.1060\\\
> \hline
> \end{array}
>   As in other clustering algorithms based on $k$-means, the proposed DRC assumes that $k$ is known. In practice, a user can compare clustering results, by varying $k$, and select the best one. This is possible since the training complexity of DRC is manageable (usually less than 3 hours). Also, we refer the readers interested in the selection of $k$ to [1].
>
>  These impacts of $k$ on clustering have been discussed in the revision. Please see the last paragraph in p. 3 and Appendix B.7 in p. 18.
>
>  [1] D.  T. Pham, S. S. Dimov, and C. D. Nguyen. Selection of k in k-means clustering. Proc. Inst. Mech. Eng. C, 2005.
>
> * **Dimensions of order and identity features:** In all experiments, we set the dimension of an order feature and an identity feature to $d_{\rm or}=128$  and $d_{\rm id}=896$, respectively. This has been specified in Appendix A in p. 14.
>
> ***
> Every attempt has been made to address your comments faithfully in the revised paper. If you have any additional comments, please let us know.
>
> Thank you again for your positive and constructive comments. We do appreciate them.

---

### Official Review · AnonReviewer3 · 2020-10-29
**The proposed method has good indicators and visualization effects. Compared with the previous methods, a better framework is proposed. The expression of the article is very clear, but some basic theories need not be explained in detail.**

**Rating:** 7
**Confidence:** 4

**Review:**

The novelty of the network structure is marginal. The decomposition way of feature is very common in computer vision. Just utilizing the latent vector of the encoder with only the comparator loss to decompose the feature into two feature types is limited. The authors should show the visual differences between these two feature types. The expression of the article is very clear, but some basic theories need not be explained in detail (Such in Section 3.4)
One more concern : h_id and h_or are both used for reconstruction. It’s best to prove that only using identity feature h_id is better than the overall latent vector h_id + h_or.

---

> ### Author Response · Authors · 2020-11-18
> **Response to reviewer 3**
>
> Thank you for your positive review and valuable comments. Please find our responses below.
> ***
>
> * **Limited feature decomposition:** As you pointed out, the proposed ORID, which employs the comparator loss and the reconstruction loss for the decomposition, does not theoretically guarantee the complete separation of order and identity information. However, in our empirical study, we found that the decomposition is sufficiently reliable.  Age transformation results in Figures 6 and 17 support this reliability. Moreover, in the revision, we have reported the accuracies of the comparator (ternary classifier), when identity features are used instead of order features. Table below lists the accuracies. The comparator fails to predict ordering relationship between two identity features accurately, which indicates that order-related information is effectively excluded from identity features. More results, including the table below, have been included to demonstrate the reliability of ORID. Please see Appendix B.9 in p. 19.
>
>     |    |\_Order feature\_|\_Identity feature\_|\_Random Guess\_|
> |---|:---:|:---:|:--:|
> |Setting A|81.4|43.2|33.3|
> |Setting B|75.4|40.8|33.3|
>
> * **Detailed explanation of basic theory:** We agree. We have shortened the explanation of the MAP decision rule in the main paper and moved the details to Appendix B.10 in p. 20.
>
> * **Reconstruction using different features & visual difference of decomposed features:** We have compared reconstruction results using the whole feature, the identity feature only, and the order feature only in Figure 7 in p. 8. Each reconstructed face using the whole feature provides significantly better quality than that using either order or identity feature only. Without the order feature, a decoded face is degraded but the person can be identified. In contrast, without the identity feature, the reconstruction is not related to the person except that it seems to be an average face of people at the same age as the person. These results confirm that order and identity features are complementary. This has been discussed in 4th paragraph in p.7. Moreover, we have provided additional t-SNE plots visualizing the feature spaces in Appendices. Please see Figure 12 in p. 17 and Figure 20 in p. 24.
>
> ***
> We have revised the paper to address all your comments faithfully. If you have any additional concerns, please let us know.
>
> Thank you again for your positive and constructive comments. We do appreciate them.

---

### Author Response · Authors · 2020-11-12
**Author response to all reviewers**

We would like to thank all reviewers for their time and thoughtful comments.

We have been carefully preparing our responses to all suggested comments, and we will upload our response to each question/comment as soon as possible.

---

### Decision · Program_Chairs · 2021-01-07
**Final Decision**

**Decision:**

Accept (Poster)

**Comment:**

This paper is overall well written and clearly presented. The problem of ordered data clustering is relevant, and the proposed method is effective.

During the discussion, all reviewers agree with the strength of this paper and share the positive impression. The authors successfully addressed reviewers' concerns by the careful author response, which I also acknowledge.
One of the reviewers raised the concern about the broader impacts, while it is also well addressed in the author response.

I therefore recommend acceptance of the paper.